# Identification of the regulatory circuit governing corneal epithelial fate determination and disease

**Jos G. A. Smits**[1], **Dulce Lima Cunha**[1], **Maryam Amini**[2], **Marina Bertolin**[3],
**Camille Laberthonnière**[1], **Jieqiong Qu**[1,4], **Nicholas Owen**[5], **Lorenz Latta**[2,6], **Berthold Seitz**[6],
**Lauriane N. Roux**[7], **Tanja Stachon**[2], **Stefano Ferrari**[3], **Mariya Moosajee**[5,8],
**Daniel Aberdam**[7,9], **Nora Szentmary**[2], **Simon J. van Heeringen**[1]\*, **Huiqing Zhou**[1,10]\*

**1** Faculty of Science, Department of Molecular Developmental Biology, Radboud Institute for Molecular Life Sciences, Radboud University, Nijmegen, the Netherlands, **2** Dr. Rolf M. Schwiete Center for Limbal Stem Cell and Aniridia Research, Saarland University, Homburg/Saar, Germany, **3** Fondazione Banca degli Occhi del Veneto, Venice, Italy, **4** Department of Medical Microbiology, Radboud University Medical Center, Radboud Institute for Molecular Life Sciences, Nijmegen, the Netherlands, **5** Development, Ageing and Disease, UCL Institute of Ophthalmology, London, United Kingdom, **6** Department of Ophthalmology, Saarland University Medical Center, UKS, Homburg, Germany, **7** INSERM U976, Paris, France, **8** Department of Genetics, Moorfields Eye Hospital NHS Foundation Trust, London, United Kingdom, **9** Université de Paris, INSERM U1138, Centre des Cordeliers, Paris, France, **10** Department of Human Genetics, Radboud University Medical Center, Nijmegen, the Netherlands

\* s.vanheeringen@science.ru.nl (SJVH); j.zhou@science.ru.nl (HZ)

**Data Availability Statement:** All cell raw sequencing files generated in this study have been deposited in the GEO database with the accession number GSE206924. Publicly accessible data was

## Abstract

The transparent corneal epithelium in the eye is maintained through the homeostasis regulated by limbal stem cells (LSCs), while the nontransparent epidermis relies on epidermal keratinocytes for renewal. Despite their cellular similarities, the precise cell fates of these two types of epithelial stem cells, which give rise to functionally distinct epithelia, remain unknown. We performed a multi-omics analysis of human LSCs from the cornea and keratinocytes from the epidermis and characterized their molecular signatures, highlighting their similarities and differences. Through gene regulatory network analyses, we identified shared and cell type-specific transcription factors (TFs) that define specific cell fates and established their regulatory hierarchy. Single-cell RNA-seq (scRNA-seq) analyses of the cornea and the epidermis confirmed these shared and cell type-specific TFs. Notably, the shared and LSC-specific TFs can cooperatively target genes associated with corneal opacity. Importantly, we discovered that *FOSL2*, a direct PAX6 target gene, is a novel candidate associated with corneal opacity, and it regulates genes implicated in corneal diseases. By characterizing molecular signatures, our study unveils the regulatory circuitry governing the LSC fate and its association with corneal opacity.

## Introduction

Cell fate determination is a complex process essential for normal development and homeostasis. The key role of transcription factors (TFs) in this process has been demonstrated by a

downloaded from GEO using the accession codes provided in S1-S3 Tables. All code used in this study is available at https://github.com/JGASmits/regulatory-networks-in-epidermal-and-corneal-epithelia, all processed epigenetic and sequencing data are available in a UCSC Genome Browser trackhub via: https://mbdata.science.ru.nl/jsmits/epi_fate/epi_fate_trackhub.hub.txt and are available on Zenodo https://zenodo.org/record/8304887.

**Funding:** This work was supported by the Aard- en Levenswetenschappen, Nederlandse Organisatie voor Wetenschappelijk Onderzoek NWO-ALW (ALWOP 376 to JGAS, HZ), the European Joint Programme Rare diseases EJP-RD/ZonMw (JPRD20-135/463003005 to DLC, HZ), ZonMw Open (09120012010039 to HZ). In addition, this study is based upon the work from COST Action CA18116, 'ANIRIDIA-NET', supported by COST (European Cooperation in Science and Technology) (to DLC, LL, SF, NS, DA, HZ). Among these fundings, JGAS received the salary from NWO-ALW (ALWOP 376) and DLC from EJP-RD/ZonMw (JPRD20-135/463003005). The funders had no role in study design, data collection and analysis, decision to publish, or preparation of the manuscript.

**Competing interests:** The authors have declared that no competing interests exist.

**Abbreviations:** BPE, bovine pituitary extract; CRE, *cis*-regulatory element; CUT&RUN, cleavage under targets and release using nuclease; EEC, ectrodactyly, ectodermal dysplasia, and cleft lip/palate; EGF, epidermal growth factor; ESC, embryonic stem cell; GO, Gene Ontology; GSEA, gene set enrichment analysis; GWAS, genome-wide association; HPO, human phenotype ontology; IP, immunoprecipitation; KC, keratinocyte; KSFM, keratinocyte serum-free medium; LSC, limbal stem cell; LSCD, limbal stem cell deficiency; qPCR, quantitative polymerase chain reaction; RT, room temperature; scRNA-seq, single-cell RNA-seq; SNV, single-nucleotide variant; TF, transcription factor; TSS, transcription start site.

plethora of seminal studies where cell conversions can be achieved by forced expression of specific sets of TFs, e.g., generation of induced pluripotent stem cells [1,2]. TFs control cell fate determination by regulating the transcriptional program, through binding to *cis*-regulatory elements (CREs) on the DNA and by modifying the chromatin environment [3,4]. This precise control is essential for tissue integrity and tissue-specific function, and deregulation often leads to pathological conditions [5,6].

The corneal epithelium in the eye and the skin epidermis are 2 types of stratified epithelia, both derived from the surface ectoderm during embryonic development. The human corneal epithelium is the outermost layer of the cornea, supported by underlying stroma and endothelium, protecting the eye from the outside environment [7–9]. It is avascular and transparent, which allows the light into the eye. The proper structure and function of the corneal epithelium are maintained by limbal stem cells (LSCs), located in the limbus, at the rim of the cornea. Differentiating LSCs move centrally to form basal epithelial cells and stratify to form differentiated epithelial layers [10]. Similar to the corneal epithelium in barrier function, the skin epidermis, on the other hand, is nontransparent. The homeostasis of the epidermis is controlled by keratinocytes (KCs) in the basal layer of the epidermis. Basal KCs differentiate vertically and outwards to form different strata of the epidermis [7]. Both LSCs and KCs are similar in their cellular morphology, even indistinguishable when cultured in vitro, and share expression of basal epithelial genes such as *KRT5* and *KRT14*. Nevertheless, their cell fates are intrinsically distinct, as they initiate and maintain specific epithelial differentiation programs that give rise to the transparent corneal epithelium and nontransparent epidermis, respectively. Insights into the comparison between cell fates of LSCs and KCs will shed light on the control mechanism of their cellular function and related pathological conditions, e.g., corneal opacity. So far, however, the cell fate similarities and differences between KCs and LSCs controlled by TFs and their associated epigenetic mechanisms are not yet understood.

In KCs and the epidermis, key TFs have been studied extensively, both in vitro and in vivo [11–14]. Key TFs include p63, GRHL family proteins, KLF4 and ZNF750, which all regulate transcriptional programs important for KC proliferation and differentiation [12,13,15,16]. Many of these TFs, often working together, are known to modulate the chromatin landscape through enhancers [12,13,16]. The TF p63 encoded by *TP63* is a key regulator of stratified epithelia and is important for the commitment, proliferation, and differentiation of KCs [13]. It binds mainly to enhancers and maintains the epigenetic landscape for the proper epidermal cell identity [16–18]. Mutations in *TP63* are associated with developmental disorders like ectrodactyly, ectodermal dysplasia, and cleft lip/palate (EEC) syndrome (OMIM 604292). In EEC, patients present defects in ectodermal derivatives, e.g., epidermis, hair follicles, and nails, but also in other epithelium-lined tissues such as the cornea [19–21]. The disease phenotypes of *TP63* mutation-associated disorders are consistent with p63 expression in stratified epithelia [19–21]. It has been shown that loss of the typical epidermal identity due to rewired epigenetic circuitry is characteristic of KCs carrying *TP63* mutations associated with EEC [16].

As compared to the wealth of molecular insights of TFs in KCs, the control mechanism of TFs in the corneal epithelium and LSCs is less understood. One of the better-studied TFs is the eye master regulator PAX6. PAX6 is essential for the specification and determination of different parts of the eye, including the retina, iris, lens, and cornea [22–24]. In the retina and lens, PAX6 interacts with chromatin modifiers such as EZH2, and cooperates with and regulates other TFs to define cell fates [25–27]. In LSCs of the cornea, PAX6 binds to enhancers, together with TFs such as RUNX1 and SMAD3, important for controlling the LSC identity [23,28–32].

Mutations and deregulation of *PAX6* are associated with aniridia (OMIM 106210), a disorder initially characterized by an absent or underdeveloped iris, among other phenotypes such

as defects in the retina, pancreas, and neurological systems [33], which is consistent with PAX6 expression in these tissues and organs [24]. Relevant to the cornea, up to 90% of aniridia patients show progressive limbal stem cell deficiency (LSCD) leading to corneal opacities [31,34]. Interestingly, LSCD and corneal opacities are also present in over 60% of patients with *TP63* mutation-associated EEC syndrome [35,36]. In addition to PAX6 and p63, another TF that has been associated with corneal abnormalities is FOXC1, of which mutations are involved in the spectrum of anterior segment dysgenesis, including Peters anomaly and Axenfeld–Rieger syndrome (OMIM 602482) [37]. FOXC1 is expressed in the epithelium, stromal, and endothelial cells of the cornea and is shown to be upstream and regulating PAX6 [38,39]. Recently, 2 reports suggested that loss of *PAX6* or *FOXC1* in LSCs gives rise to loss of the LSC identity, and these PAX6 or FOXC1-deficient LSCs acquire a KC-like cell signature, indicated by up-regulated expression of epidermal stratification marker genes [30,39,40]. How TFs like PAX6, p63, and FOXC1 regulate their target genes in LSCs and how their mutations give rise to LSCD and corneal opacities are not yet fully understood. Therefore, a comprehensive characterization and comparison of molecular signatures between LSCs and KCs will not only identify shared and tissue-specific TFs controlling cell fates but also provide insights into the pathomechanisms of LSCD and other corneal opacity disease mechanisms.

In this study, we performed in-depth analyses of the transcriptome and the epigenome of human LSCs and KCs cultured in vitro and characterized differentially expressed genes and regulatory regions between the 2 cell types. Using a gene regulatory network-based method, we identified key TFs and their hierarchy controlling epithelial programs that are shared by KCs and LSCs and those that are distinct for each cell type. Expression patterns of the key TFs were further validated with in vivo single-cell RNA-seq (scRNA-seq) data from the cornea and the epidermis. Importantly, we showed that the key TFs and their target genes that drive the LSC-specific epithelial program are associated with corneal diseases and identified a novel disease candidate *FOSL2* associated with corneal opacity.

## Results

### Distinct epithelial gene expression patterns define cell fate differences of skin keratinocytes and cornea limbal stem cells

To characterize gene expression patterns that define the cell fate difference between human cornea LSCs and human skin KCs (Fig 1A), we used cultured LSCs established from post mortem limbal biopsies and basal KCs from skin donors. Both cultured cells have the capacity to regenerate stratified epithelial tissues in vitro [41,42] and have high p63 expression (S1 Fig), thus exhibiting the progenitor cell state. We performed comparative RNA-seq analyses from bulk and pseudobulk RNA-seq data (aggregated from scRNA-seq using cultured LSCs and KCs) experiments and incorporated our data with publicly available RNA-seq data for robustness (S1 Table) (S1 Fig) [16,32]. Single-cell data was aggregated because no measurable heterogeneity was detected in these cultured cells, except for cell cycle differences (S1E Fig).

Through pairwise comparison, we identified 1,251 differential expressed genes between LSCs and KCs. Among them, 793 genes had higher expression in LSCs (referred to as LSC-high genes), while 459 differential genes were more highly expressed in KCs (referred to as KC-high genes). This analysis resulted in typical genes for both epithelial cell types: LSC-high genes with limbal and corneal epithelial genes including KRT19 and the eye master regulator PAX6 (Figs 1B and S1F), whereas KC-high genes contained epidermal markers such as *KRT1*, *KRT10*, *LCE3D*, and *LCE3C*. Although some of these detected genes are associated with epithelial stratification, e.g., *KRT3* and *KRT12* for the cornea and *KRT1* and *KRT10* for the epidermis, their expression was significantly lower (5- to 20-fold) than their expression in stratified

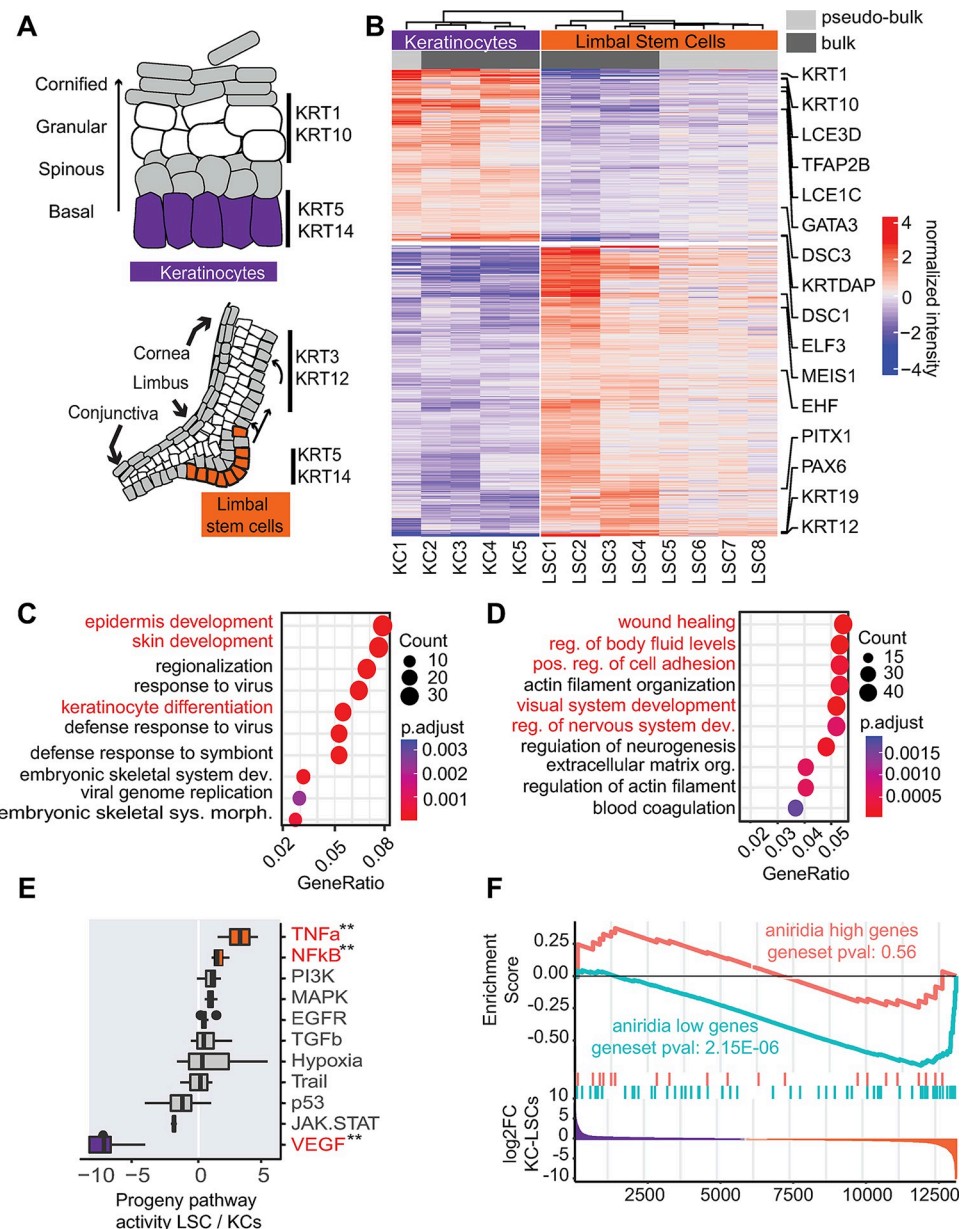

**Fig 1. RNA-seq analysis of LSCs and KCs.** (A) Schematic picture of the epidermis and the limbus. (B) Heatmap of normalized expression of differentially expressed genes between LSCs and KCs (adjusted pval < 0.01, log2 FC > 1.5). Differentially expressed genes are clustered using k-means clustering with 2 clusters. (C) GO term enrichment of KC-high genes. (D) GO term enrichment of LSC-high genes. (E) PROGENy pathway activity analysis, with scores sorted based on the LSC/KC ratio. Pathways depicted in red are differential, the color is gray if non-differential, orange if higher in LSC, and purple if higher in KCs. (F) GSEA of differentially expressed genes identified in aniridia patient LSCs, as compared to controls, up- and down-regulated genes (aniridia high and low, respectively) were tested for enrichments against the KC-LSC fold change. For all underlying data, see S4 Table, GEO GSE206922, GSE206923, and GSE242995. GO, Gene Ontology; GSEA, gene set enrichment analysis; KC, keratinocyte; LSC, limbal stem cell.

epithelial cells [16,39] (S1G Fig), confirming the progenitor states of the cultured LSCs and KCs. It should be noted that TP63 is highly expressed in both LSCs and KCs (S1F Fig), and therefore is not identified as differential.

Gene Ontology (GO) enrichment analysis [43,44] of KC-high genes identified enrichment of GO terms related to the "epidermis" and "skin development" (Fig 1C). GO terms associated with "response to virus" were enriched due to detected immune and interferon-related genes among KC-high genes, consistent with the gene set enrichment analysis (GSEA) using the hallmark gene set of the MsigDB collection [45,46] that identified enrichment of the interferon-alpha and gamma response (S4 Table). Finally, PROGENy pathway target gene analysis [47] identified higher VEGF signaling in KCs (Fig 1E), indicating that VEGF vascularization-related genes are lowly expressed or completely repressed in LSCs. This is in line with the avascularized state of the cornea. LSC-high genes were enriched for terms such as "wound healing" and "positive regulation of body fluid levels" as well as "visual system development" and "regulation of nervous system development" (Fig 1D). PROGENy analysis identified the TNF-α and NFKβ pathways associated with LSCs (Fig 1E), such as *CXCL1,3,5,6* and *TNF-α*. Consistently, TNF-α and NF-kB signaling was also identified by KEGG pathway [48], GO, and GSEA analyses using the hallmark gene set of the MsigDB collection (S2 Fig). Finally, GSEA enrichment using the C8 dataset of the MsigDB collection that contains single-cell dataset gene lists also identified enrichment for genes in the "Descartes fetal eye corneal and conjunctival epithelial cells" (S4 Table) with more signal in the LSCs.

Next, we asked the question of whether LSCD-associated LSCs acquire a KC-like cell fate. To test this, we examined the cell fate of LSCs from patients with aniridia, a disease mostly caused by *PAX6* haploinsufficiency. We performed RNA-seq analysis of primary LSCs of 2 aniridia patients and postmortem cornea-extracted LSC controls. Aniridia and control LSCs were both expanded in keratinocyte serum-free medium (KSFM) conditions and were on passage 3 when processed. These data were integrated with other aniridia RNA-seq data published previously using the same procedures and culture conditions [49] (S1 Table). We obtained 73 differentially down-regulated genes (aniridia low genes) and 22 up-regulated genes (aniridia high genes) in aniridia patient LSCs, as compared to LSCs from healthy controls (S3 Fig). Aniridia low genes included the *PAX6* target gene *KRT12* and other corneal and epithelial genes such as *TGFBI*, *CLND1*, *GJB6*, *IL36G*, *LAYN*, *NMU*, and *TMEM47*. Many of these epithelial genes are potential PAX6 target genes reported in an immortalized LSC model where 1 allele of *PAX6* was deleted [50]. We then applied GSEA to compare them to LSC and KC gene expression signatures, to investigate whether these deregulated genes due to *PAX6* haploinsufficiency represent the changed cell fate of these aniridia LSCs. Indeed, as expected, aniridia low genes were enriched among genes expressed highly in LSCs (*P*-value 0.000028) (Fig 1F), indicating a loss of the LSC cell fate in aniridia LSCs. Among aniridia high genes, we found *GATA3* present in genes expressed highly in KCs (S3 Fig). Nevertheless, there was no detected enrichment of aniridia high genes among genes expressed highly in KCs representing the KC cell fate, arguing against the postulated model that PAX6-deficient LSCs acquire a KC-like cell fate at the transcriptome level. These data suggest that additional mechanisms, such as TFs other than PAX6, contribute to the cell fate difference between KCs and LSCs.

## The epigenetic states of *cis*-regulatory elements correlate with gene expression patterns

To understand the mechanisms underlying different cell fate controls of LSCs and KCs, we identified CREs and their epigenetic states that drive gene expression differences. We generated an extensive multi-omics dataset of LSCs and KCs and integrated these with other published data (S4A Fig and S2 and S3 Tables).

The complete dataset included ATAC-seq for open chromatin regions representing CREs [32] and ChIP-seq of histone modifications, H3K27ac and H3K4me3 marking active CREs,

and H3K27me3 that marks repressed chromatin regions (Fig 2A) [17,32]. Using ATAC-seq analysis, we identified 124,062 CREs in the 2 cell types. Approximately 80% of these CREs were accessible in both cell types (S5 Fig). The public ATAC-seq data was compared to the data generated using Pearson correlation (S5F Fig). This comparison revealed a distinct biological difference between KCs and LSCs, which was more prominent than technical variation across different laboratories and techniques.

To examine the differential epigenetic states of these CREs in LSCs and KCs, we quantified ATAC-seq and histone modification signals in windows covering these CREs (Fig 2A). This resulted in 35,348 CREs with differential epigenetic signals, about 28.5% of CREs (S5 Fig). To assess the correlation between these differential CREs and the expression of their nearby genes, we considered both CREs at the promoter regions (promoter CREs) and enhancer CREs located within 50-kb distance from the genes (enhancer CREs) (Figs 2B and S4B). As expected, high ATAC, H3K4me3, and H3K27ac signals correlated with high gene expression, while the strong signals of repressive H3K27me3 correlated well with lowly expressed genes or genes with undetectable expression in the corresponding cell types, e.g., the loci of *PAX6*, *GATA3*, *HOXA9*, and *TNF-α* (Figs 2B, 2D and S4D). The strong repression signals marked by H3K27me3 in KCs at genes that are key for the LSC fate such as *PAX6* suggest a repression mechanism in KCs to prevent inappropriate gene expression that defines the LSC fate. All annotated CRE regions are available on a publicly accessible trackhub [51] (see Data availability).

Next, we performed GO analysis on genes that are close to differential CREs (Fig 2C). For H3K27ac and H3K4me3 that mark active CREs, GO terms such as "epidermis" and "skin development" were identified for CREs with strong signals in KCs. Furthermore, "positive regulation of cell adhesion" and "extracellular matrix organization" terms were found for CREs with strong H3K27ac and H3K4me3 signals in LSCs. This is consistent with identified differentially expressed genes in each corresponding cell type. In contrast, the repressive mark H3K27me3 is anti-correlated with gene expression (Fig 2C).

Furthermore, in line with the enrichment of *TNF-α* and *NF-kB* signaling pathways in LSCs identified by PROGENy analysis of differentially expressed genes, higher H3K27ac and H3K4me3 signals were present in the loci of *TNF-α* and *NF-kB* target genes in LSCs, as compared to in KCs, while these loci in KCs were repressed by H3K27me3 (S4C Fig).

## Gene regulatory network analysis identifies transcription factors controlling distinct epithelial cell fates and their hierarchy

Using the identified differential CREs, we set out to identify key TFs driving the cell fate differences between LSCs and KCs. TF-binding motif enrichment was performed using Gimme Motifs [52] in all differential CREs marked by ATAC, H3K4me3, H3K27ac, and/or H3K27me3 signals. In general, TF motifs enriched in CREs with active marks, ATAC, H3K4me3, or H3K27ac, in 1 cell type, were also enriched within regions with the repressive H3K27me3 mark in the other cell type (Figs 3A and S6A). For example, TF motifs that are linked to FOXC1, TEAD1, JUN, PAX6, FOS, RUNX2, OTX1, ELF3, SOX9, and REL were detected in differential CREs marked by high active mark signals in LSCs but also marked by high H3K27me3 in KCs. Consistent with our expectation, the enrichment of the PAX6 motif in differential CREs with higher active mark signals was detected in LSCs, as PAX6 is specific for LSCs but not for KCs. As REL is a TF involved in TNF-α and NF-kB pathways, the detection of the REL motif is consistent with the enrichment of TNF-α and NFKβ signaling genes among LSC-high genes. Notably, the FOS motif that is associated with FOS, FOSL1, FOSL2, and JUN (S6A Fig) was present in approximately 10% of all variable CREs in LSCs, the highest

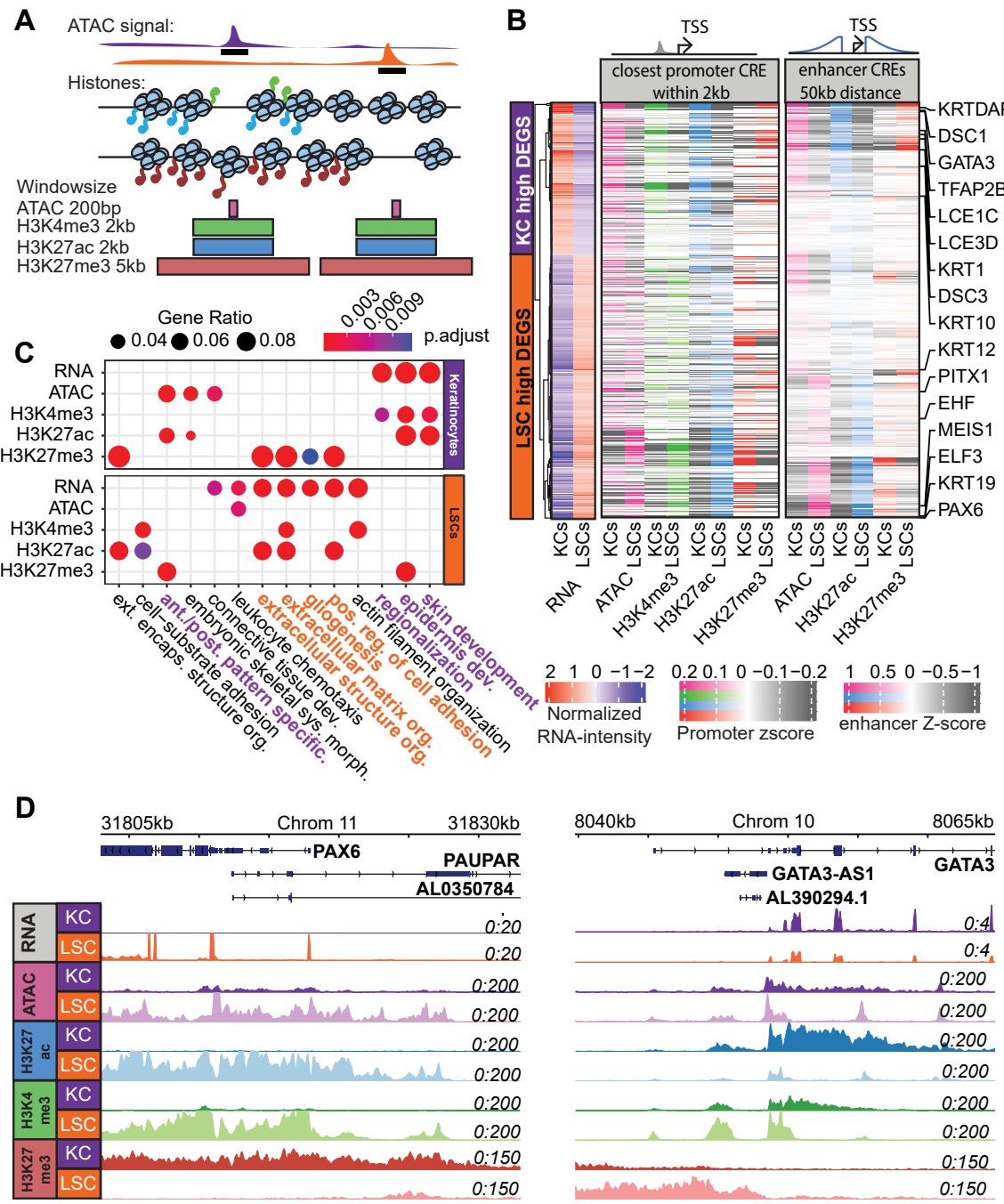

**Fig 2. CRE analysis.** (A) Schematic overview of CRE identification and quantification. Signals of each analysis were quantified by different window sizes covering the ATAC-seq peak summit. (B) Heatmap of the Z-scores of the quantile normalized ATAC-seq and histone mark signals near LSC- and KC-high genes. For promoter CREs, it corresponds to the closest CRE within 20 kb to the TSS. For enhancer CREs, the signals of all CREs within a 100-kb window near a TSS were quantified, distance weighted, and summed. (C) GO term enrichment of LSC- and KC-high genes and genes close (within 20 kb) to differential CREs. (D) PAX6 and GATA4 TSS loci with signals of RNA-seq, ATAC-seq, ChIP-seq of H3K27ac, H3K4me3, and H3K27me3 in KCs and LSCs. For the underlying data, see S5 Table, GEO GSE206918, GSE206920, and the trackhub in the Zenodo entry [51]. CRE, *cis*-regulatory element; KC, keratinocyte; LSC, limbal stem cell; TSS, transcription start site.

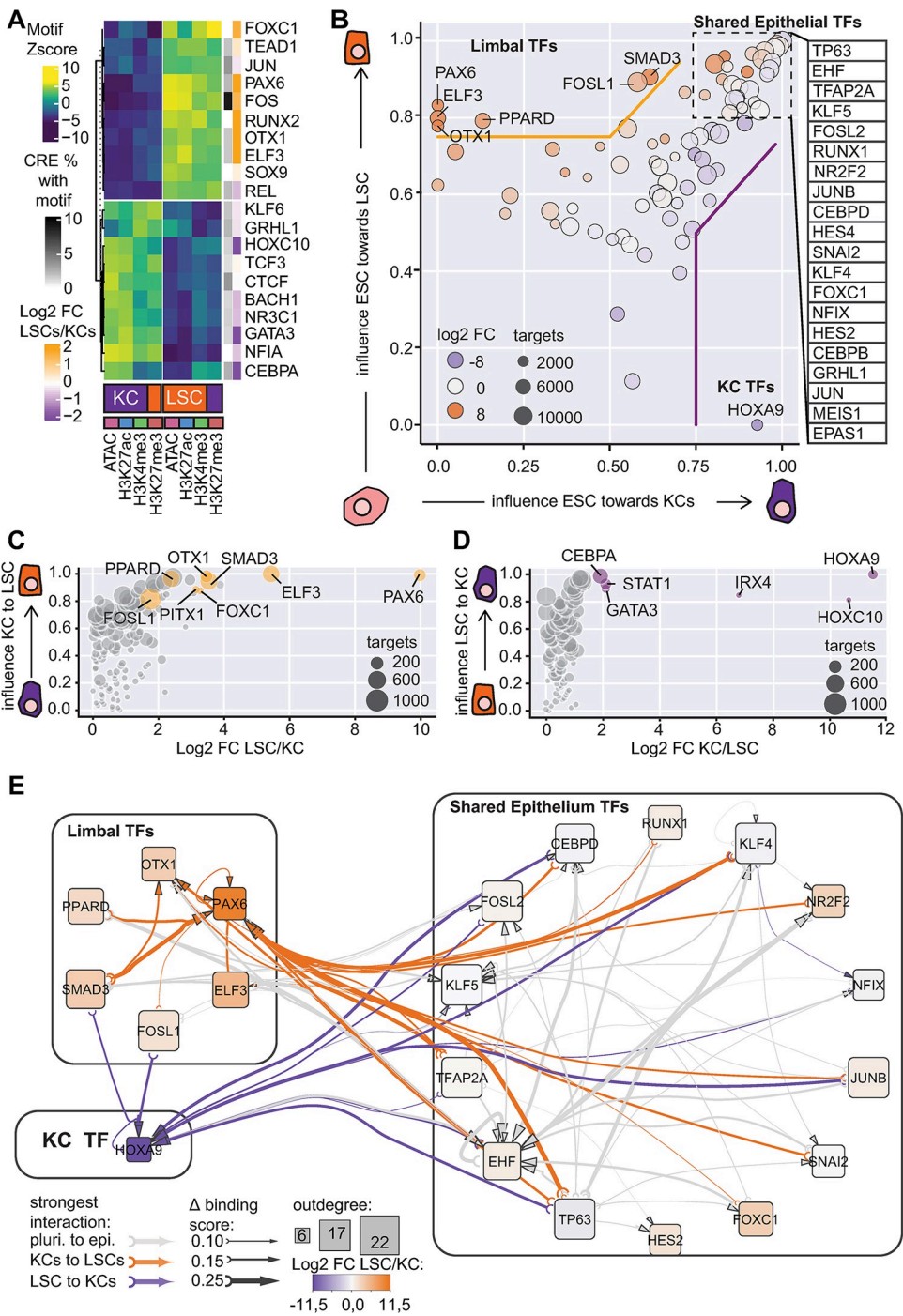

**Fig 3. TFs and TF hierarchy controlling distinct epithelial cell identity.** (A) Heatmap of motif enrichment Z-scores detected in variable CREs and the corresponding TFs. The percentage of CREs containing the motifs and the expression ratio of TFs in LSCs and KCs are indicated. (B) ANANSE influence score plot of TFs identified in ESC-KC (x-axis) and ESC-LSC (y-axis) comparison. Circle size represents the maximum number of target genes of a TF. The color represents log2FC between LSC/KC (orange LSC high; purple KC high). (C) ANANSE influence score plot of TFs identified in KC-LSC comparison. (D) ANANSE influence score plot of TFs identified in LSC-KC comparison. (E) TF hierarchy is indicated by the binding score of a TF to its target TF locus, and the cell type-specific regulation is indicated by the binding score difference of the TF at the target TF locus between cell types. When a binding score difference in KC-LSC comparison is greater than the mean of the difference in ESC-KC and ESC-LSC comparison, this TF regulation of the target TF is annotated as either KC- (purple arrows) or LSC-specific (orange arrows) regulation. Otherwise, the regulation is annotated as "shared regulation" for both cell types (gray arrows). The degree

of binding score difference is indicated by the thickness of the arrows. Outdegree node size represents the number of target genes. Fold change of TF gene expression in LSC and KCs is represented by orange (LSC-high) and purple (KC-high) colors. For the underlying data, see the Zenodo entry [51]. CRE, *cis*-regulatory element; ESC, embryonic stem cell; KC, keratinocyte; LSC, limbal stem cell; TF, transcription factor.

among all motifs (Fig 3A). Motifs enriched in KC active CREs included those linked to KLF6, GRHL1, HOXC10, GATA3, NFIA, CEBPA, and CTCF. These motifs were also enriched in CREs marked by high H3K27me3 signals in LSCs. Enriched motifs could mostly be linked to TFs with high expression differences between LSCs and KCs, e.g., FOXC1, PAX6, and FOS are highly expressed in LSCs, while HOXC10, GATA3, and CEBPA are highly expressed in KCs (Fig 3A).

To predict gene regulation by considering the expression of both TFs and their targets, we applied ANANSE [53], a gene regulatory network method to identify key TFs for cell identities. ANANSE integrates CRE activities and TF motif predictions with the expression of TFs and their target genes to generate a gene regulatory network of the specific cell type. Subsequently, a pairwise comparison of gene regulatory networks from 2 cell types is performed to identify the most influential TFs that differentiate between the 2 cell types. The overall importance of identified TFs is represented by their influence score, scaled from 0 to 1 (Materials and methods) [53].

Because many key TFs for LSC and KC fates that are shared in these 2 cell types such as p63 had similar gene expression levels (S1F Fig), these TFs could not be identified through a direct comparison of the differential gene regulatory network between these 2 cell types using ANANSE. Therefore, we decided to utilize embryonic stem cells (ESCs), a completely different cell type as compared to either LSCs or KCs, as the reference point for this differential network analysis. This enabled us to identify not only distinct but also shared TFs for LSC and KC fates. Using RNA-seq, ATAC-seq, and H3K27ac data from ESCs [54], we performed a pairwise comparison of gene regulatory networks between ESCs versus KCs or ESCs versus LSCs. When predicting the TFs driving the LSC or KC fates from ESCs, ANANSE resulted in 70 epithelial TFs that had influence scores above 0.5 in both ESC-LSC and ESC-KC pairwise differential network analysis. Many TFs are known to be important for epithelial cell function, such as TP63, EHF, TFAP2A, TFAP2C, FOSL2, the KLF family protein 3,4,5,6,7, JUNB, CEBPD, CEBPB, and RUNX1. We classified these TFs as shared epithelial TFs (Figs 3B and S7). Intriguingly, when conducting a network outdegree analysis that quantifies the number of targets of a TF within the top 20 TFs, FOSL2, JUN, TP63, and TFAP2A were identified as regulating most other TFs. This implies their importance in driving other TF expression (S7B and S7C Fig). The highest outdegree of FOSL2 is in line with the high percentage of detected FOS motif (Fig 3A).

Importantly, in the ESC-LSC and ESC-KC differential network analysis, TFs with high influence scores in LSCs but with undetectable (PAX6, ELF3, OTX1, PPARD) or low (FOSL1 and SMAD3) influence scores in KCs were considered as LSC-specific TFs (Fig 3B). Direct pairwise differential network analysis between LSCs and KCs (Fig 3C) was largely consistent with these findings. Interestingly, FOXC1 was annotated as a shared epithelial TF, whereas in the direct KC-LSC comparison, it was identified as a specific TF for the LSC fate. This is probably due to the higher expression of FOXC1 in LSCs. For KC-specific TFs, we only identified HOXA9 in the ESC-LSC and ESC-KC differential network analysis (Fig 3B). The direct pairwise differential network analysis between KCs and LSCs (Fig 3D) confirmed this. Next to HOXA9, the direct comparison identified other TFs such as GATA3, IRX4, and CEBPA with high influence scores in KCs, indicating that KC-LSC pairwise comparison is more sensitive for detecting KC-specific TFs.

Finally, we set out to dissect the TF regulatory hierarchy for the cell identity differences between LSCs and KCs, by identifying potential target TFs of the shared and specific TFs. For this analysis, we did not consider the expression level of TFs themselves, to ensure that the prediction is mainly driven by the potential binding of a TF to its target loci, represented by the binding score. If a target TF is regulated by a TF with similar binding scores in both LSCs and KCs compared to ESCs, this regulation is annotated as a "shared regulation"; if the binding score is significantly higher in one cell type than in the other (Materials and methods), the regulation of the TF-target TF pair is annotated as "cell type-specific regulation." We included the top shared and specific TFs, 15 shared epithelial TFs, 6 LSC specific TFs (PAX6, ELF3, OTX1, PPARD, SMAD3, and FOSL1), and the KC-specific TF HOXA9 (Figs 3B, 3E, S6B, S6C and S6D). As expected, many shared TFs are regulating each other via "shared regulation" (Fig 3E, gray arrows). Consistently, cell type-specific TFs regulate their target TFs via "cell type-specific regulation" (Fig 3E, orange arrows), e.g., *PAX6* is predicted to be regulated by SMAD3 and PPARD. Furthermore, many autoregulation loops were also detected, e.g., PAX6 in LSCs and HOXA9 in KCs. We also found that shared TFs may regulate cell type-specific TFs via "cell type-specific regulation." For example, p63, FOXC1, and TFAP2A were identified as shared TFs between KCs and LSCs, but they were predicted to regulate *PAX6* in LSCs.

In summary, our molecular characterization using KCs and LSCs cultured in vitro identified shared and cell type-specific TFs for the LSC and KC fates. p63, FOSL2, EHF, TFAP2A, KLF5, RUNX1, CEBPD, and FOXC1 are among the shared epithelial TFs for both LSCs and KCs. PAX6, SMAD3, OTX1, ELF3, and PPARD are LSC specific TFs for the LSC fate, and HOXA9, IRX4, CEBPA, and GATA3 were identified as KC-specific TFs. Furthermore, LSC and KC fates are defined by cooperative regulation of both shared and cell type-specific TFs.

## Single-cell RNA-seq analysis of the cornea and the epidermis validates the expression of key transcription factors controlling cell fates

Since our multi-omics analysis was performed on LSCs and KCs cultured in vitro, we assessed scRNA-seq datasets derived from the cornea and the epidermis to confirm that the molecular signatures of LSCs and KCs in our study indeed represent those of epithelial stem cells maintaining the corneal limbus and the epidermis [55,56]. By clustering single cells according to marker gene expression (S8 Fig), we selected the cell clusters corresponding to the stem cells as pseudobulk for further differential gene expression analysis. For the epidermis, we selected cells with high *KRT14*, *KRT5*, and low *KRT1* and *KRT10* expression as basal KCs, and for the cornea, cells with high *S100A2*, *PAX6*, and *TP63* expression and without *CPVL* expression as LSCs, because *CPVL* has been proposed as a marker with neural crest origin [56].

Consistent with the in vitro findings, the in vivo LSCs expressed high levels of *PAX6*, *ELF3*, *FOXC1*, *FOSL1*, *OTX1*, and *SMAD3*, whereas the in vivo KCs expressed high levels of *HOXA9*, *CEBPA*, and *GATA3* (Fig 4A). GO analysis identified similar functions of differentially expressed genes between the in vivo LSCs and KCs, as compared to those from in vitro cultured cells (S8E and S8F Fig). Furthermore, PROGENy analysis of differentially expressed genes between in vivo LSCs and KCs showed that TNF-α and NF-kB pathway genes are significantly enriched in in vivo LSCs, e.g., *CXCL1,2,3,8,20* and *NFKB1* (Fig 4B). GSEA analysis using the hallmark gene set of the MsigDB collection also identified enrichment for TNF-α signaling genes (S5 Table).

As the data of CREs of in vivo tissues were not available, we utilized the in vitro ATAC and H3K27ac datasets for differential gene regulatory network analysis, together with the in vivo pseudobulk data used for gene expression analysis above. Here, we assumed that, since the GRN analysis is largely driven by gene expression data, this analysis assessed the influence of TFs on in vivo LSC and KC fate differences. Overall, the in vivo data identified similar cell

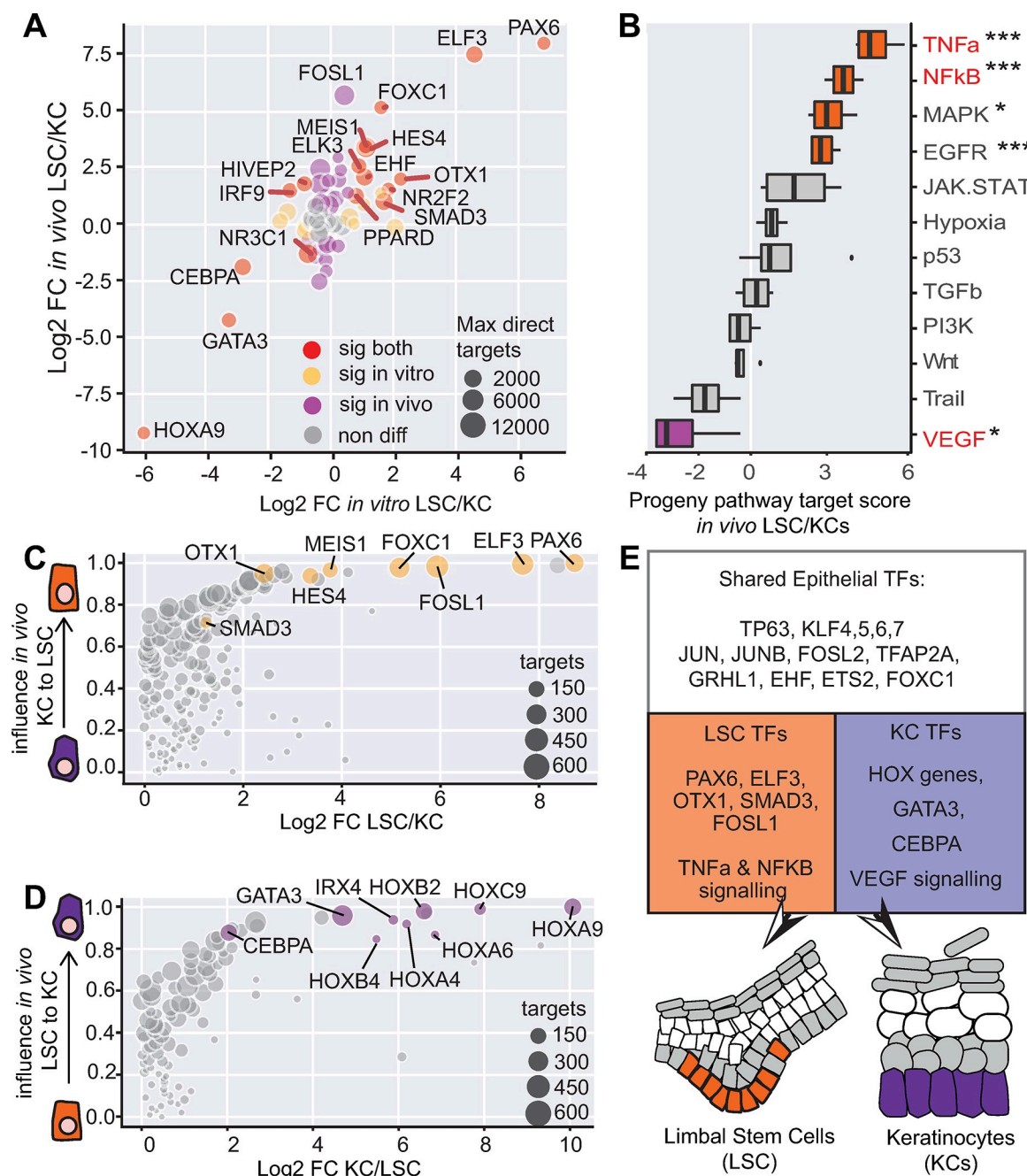

**Fig 4. Validation of key TF expression using in vivo scRNA-seq.** (A) Fold change comparison of identified TFs using in vivo and in vitro data. (B) PROGENy pathway analysis of in vivo LSCs and KCs. (C) ANANSE influence score plot of in vivo basal KCs to LSCs. (D) ANANSE influence score plot of in vivo LSC to basal KCs. (E) Summary of the identified shared and specific TFs. For the underlying data, see S6 Table, GEO GSE155683, GSE147482 [55,56], and the Zenodo entry [51]. KC, keratinocyte; LSC, limbal stem cell; scRNA-seq, single-cell RNA-seq; TF, transcription factor.

type-specific TFs, as compared to in vitro cultured cells (Fig 4C and 4D). In in vivo LSCs, PPARD however was not detected, PAX6, ELF3 FOXC1, and FOSL1 exhibited the highest influence scores, and TFs with the highest influence scores in in vivo basal KCs were HOX TFs and a few others such as CEBPA, GATA3, and IRX4.

Taken together, our analyses showed clear consistency between in vivo and in vitro derived data and identified key TFs driving the cell fate of between LSCs and KCs (Fig 4E).

## FOSL2 is a novel transcription factor controlling the LSC fate and associated with corneal opacity

As several key TFs defining the LSC fate, either shared or cell type specific, such as p63 and PAX6, respectively, are associated with corneal opacity, we explored the potential involvement of LSC TFs in the pathomechanism of corneal diseases. For this, we leveraged the whole genome sequencing data in the 100,000 Genomes Project at Genomics England UK to identify variants of uncertain significance that may have functional consequences. For establishing a suitable cohort, we identified a total number of 33 unsolved participants with human pheno-type ontology (HPO) terms associated with corneal opacity (S8 Table). Next, we screened for variants in the coding regions of the top 15 shared epithelial TFs as well as the 6 LSC-specific TFs identified in our study. In a proband with band keratopathy (HP:0000585), we identified a rare de novo heterozygous missense variant in *FOSL2* (2:28412095:C:T, genome build GRCh38/hg38, NM_005253.4:c.628C>T) (genomAD allele frequency 0.0000922) (S9A Fig), giving rise to a predicted damaging amino acid change (NP_005244.1:p.(Arg210Cys)), based on most major prediction tools (S9A Fig and S9 Table).

As the role of FOSL2 in LSCs is completely unknown, we first investigated the protein expression of FOSL2 within the cornea. The immunostaining results showed that FOSL2 protein is expressed in both limbus and central cornea (Figs 5A and S9B), co-localizing in the nucleus with other LSC TFs, e.g., p63, PAX6, and its closely related AP1 complex factor FOSL1 (S9C–S9F Fig). The same was also visible in LSCs (and KCs, with the exception of PAX6) cultured in vitro (S9G–S9J Fig). The identified variant associated with corneal opacity, together with the expression pattern, indicated an important function of FOSL2 in the cornea.

To further explore the association of FOSL2 with corneal diseases, we questioned whether the target genes of FOSL2 are enriched for corneal disease genes, similar to other known LSC TFs such as PAX6, FOXC1, and p63. The rationale behind this question is based on the concept that many disease-associated TFs could contribute to the diseases not only because of their mutations but also due to abnormal regulation of their target genes [29]. Here, our analyses investigated a broader scope of corneal diseases, because many of these known disease-associated TFs can influence multiple tissues of the eye. For this analysis, we employed 2 types of data, DNA-binding profiles of key LSC TFs to predict their target genes and a comprehensive list of corneal disease genes. For TF DNA-binding profiles, we generated cleavage under targets and release using nuclease (CUT&RUN) of FOSL2 and ChIP-seq of the p63 protein in LSCs and additionally incorporated publicly available ChIP-seq data of PAX6, FOXC1, RUNX1, and SMAD3 in LSCs [30,32]. Except for FOSL2, all other TFs are known to regulate the LSC fate. To collect genes that are associated with corneal phenotypes, we used 2 approaches. First, we performed a literature search and constructed a list of 161 genes associated with LSCD, inherited corneal opacification diseases, and ocular as well as systemic syndromes with known corneal manifestations (S7 Table). In addition to this curated disease gene list, we used disease genes assembled in the EyeDiseases database [57]. To assess whether TF binding of the key TFs to corneal disease gene loci is more probable than random, we mapped TF binding sites to the nearest genes in order to detect a statistically significant enrichment of disease genes bound by TFs (S10A Fig). With this method, we identified that the genes associated to our "curated corneal disease list" were significantly more likely to be bound by PAX6, FOXC1, RUNX1, and FOSL2 (Figs 5B and S10C). The binding signal of these TFs within the transcription start site (TSS) regions of the top cornea disease-related genes is summarized by the quantification of binding

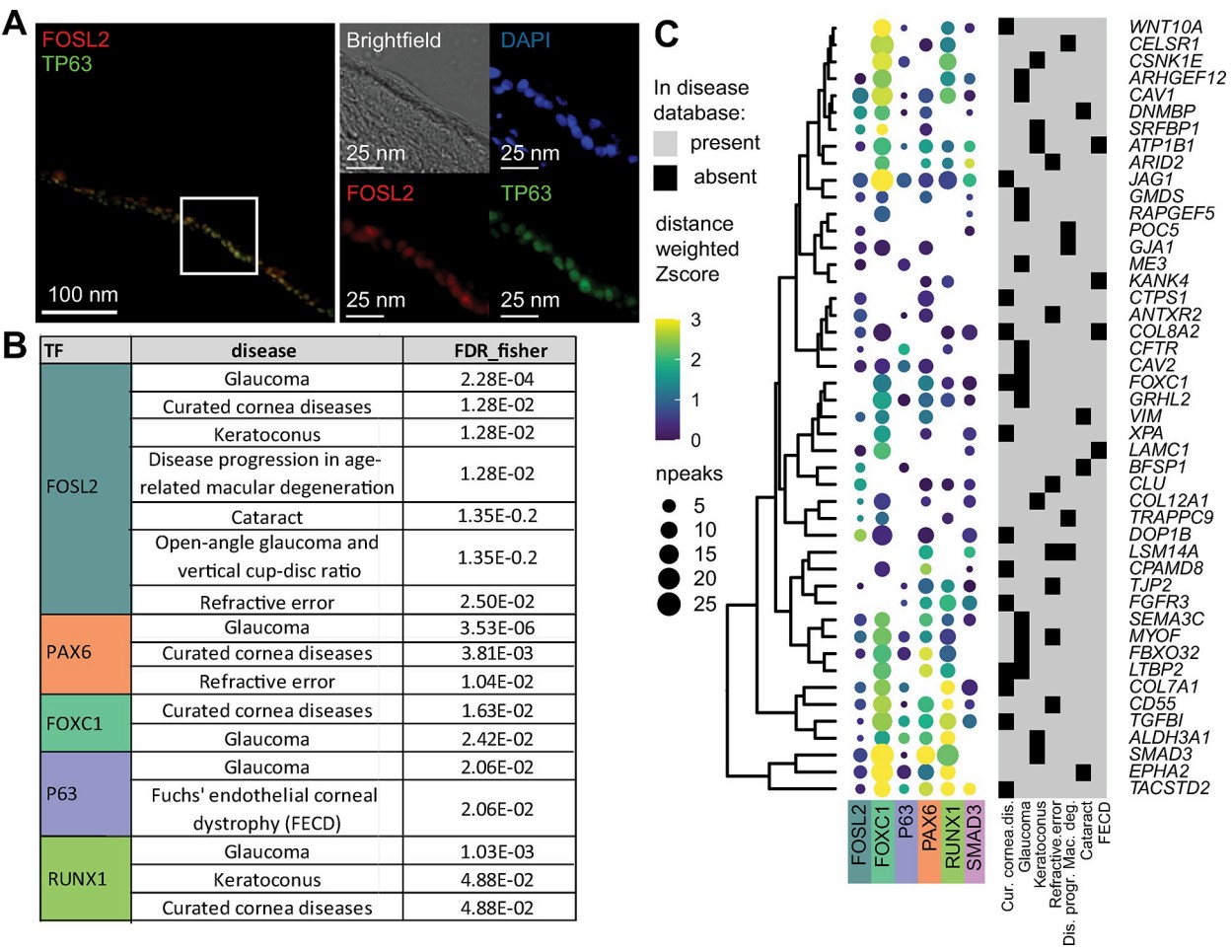

**Fig 5. Transcription factors regulation of corneal disease-associated genes.** (A) FOSL2 and p63 staining of the peripheral cornea. (B) TFs that bind to gene loci associated with corneal abnormalities with significantly higher occurrence (FDR), as compared to binding to all genes in the whole genome. FDR was calculated with Fisher exact testing. (C) Dot plot showing the TF binding intensity (color bar) and the number of binding peaks (npeaks, dot size) near the disease genes that contain a significant number of potential TF binding. The number of peaks is within a 100 kb region of the TSS; TF binding intensity score is the weighted z score of the quantile log normalized intensities distance weighted. FECD, Fuchs Endothelial Corneal Dystrophy. For underlying data see S7 Table, GEO GSE206920, GSE236440, and GSE156272. TF, transcription factor; TSS, transcription start site.

signals (Fig 5C). In parallel, we investigated the TF-disease association by combining TF binding signals with the distance between the TF binding peaks and the disease genes and employed a Mann–Whitney U statistical test to identify a potential binding increase (S10B Fig). This gave rise to similar TF-disease association enrichment results (S10C Fig).

Overall, our analysis highlighted that FOSL2 and several other key LSC TFs binding is enriched in the TSS regions of cornea disease genes, suggesting that they directly regulate cornea disease genes.

## FOSL2 is a direct PAX6 target gene and regulates angiogenesis and tight junction genes in LSCs

To further assess the function of FOSL2 in LSCs and corneal opacity, we performed siRNA knockdown of *FOSL2* in primary LSCs. Knockdown of *FOSL2* was validated using quantitative polymerase chain reaction (qPCR) and RNA-seq and displayed an efficiency of over 80%, as

compared to control siCTR (0.12-fold ± 0.068, *n* = 4) (Figs 6A, S11A and S11B), and gave rise to 212 differentially expressed genes. Because PAX6 is an undisputed regulator in LSCs and defects are associated with LSCD and corneal opacity, we also performed *PAX6* knockdown (siPAX6). Intriguingly, *FOSL2* expression was significantly down-regulated upon *PAX6* knockdown, as compared to siCTR (0.5-fold) (Fig 6A). Furthermore, several high-confidence PAX6 binding peaks were identified surrounding the *FOSL2* locus (Fig 6B), demonstrating that *FOSL2* is a direct target gene of PAX6 in LSCs.

Among the 212 differentially regulated genes detected in siFOSL2 (Fig 6C), 108 were up-regulated, with significant enrichment of genes involved in "negative regulation of cell population proliferation," "blood vessel morphogenesis," "angiogenesis," and "endothelial cell differentiation" (Fig 6D). Furthermore, predicted FOSL2 target genes based on the nearby FOSL2 binding sites were enriched among up-regulated genes by siFOSL2 (Fig 6E). This indicates that FOSL2 may function primarily as a repressive factor in LSCs. Relevant up-regulated genes included *TGFBI* and *JAG1* which have known roles in angiogenesis and *CLDN1* which is involved in tight junctions (Fig 6F). Furthermore, FOSL2 binding signals were identified near these genes (Fig 6G), indicating that these genes are direct targets of FOSL2.

Interestingly, 103 down-regulated genes upon si*FOSL2* did not give any enrichment in GO analysis, suggesting a less prominent role of FOSL2 in gene activation. Nevertheless, *CLDN4* and *CLDN7*, other tight junction genes, are significantly down-regulated (Figs 6C, 6F and S11G), suggesting an altered barrier function in FOSL2-deficient LSCs. Other down-regulated genes include *TGM1* and *ABCA12*, linked to epidermal permeability barrier disorders (S11G Fig).

To summarize, our results demonstrated that, among TFs that define the LSC fate, FOSL2 is a novel LSC TF associated with cornea opacity.

## Discussion

The corneal epithelium and the epidermis are both stratified epithelia, serving as barriers and the first-line defense against external insults. Nevertheless, they have distinct tissue-specific functions that are tightly controlled by the proliferation and differentiation program of their corresponding stem cells, LSCs for the cornea and basal KCs for the epidermis. In this study, we characterized molecular signatures defining the cell fates of these 2 cell types by integrating in-house multi-omics data as well as publicly available datasets. Using motif and gene regulatory network analyses, we identified a collection of shared and cell type-specific epithelial TFs defining KCs and LSCs. Furthermore, we showed a proof-of-principle that this resource of LSC TFs and their regulatory mechanisms can provide novel tools for dissecting pathomechanisms of corneal diseases.

In contrast to the well-studied TFs and their associated gene regulatory networks in epidermal KCs [11–14,58], TFs regulating cornea LSCs have only recently started to emerge. Except for PAX6, identified TFs including p63, SMAD3, RUNX1, and FOXC1 [30,32,39] which are all also expressed in KCs [11–14,58]. This raises interesting questions about whether these TFs are sufficient to determine cell fate differences between LSCs and KCs and how they control cell fate determination mechanisms. By specifically comparing LSCs to KCs, we identified PAX6, SMAD3, OTX1, ELF3, and FOSL1 as the LSC-specific TFs that determine the LSC fate. To note, these LSC-specific TFs may be good candidates for developing (trans)differentiation strategies to generate LSCs from other cell types for corneal regenerations [59,60]. The identification of PAX6 as an LSC-specific TF was expected. It is an eye development master regulator [30] and is associated with the disease aniridia where corneal opacity is one of the main manifestations [33]. Furthermore, PAX6 has previously been shown to co-regulate target genes with

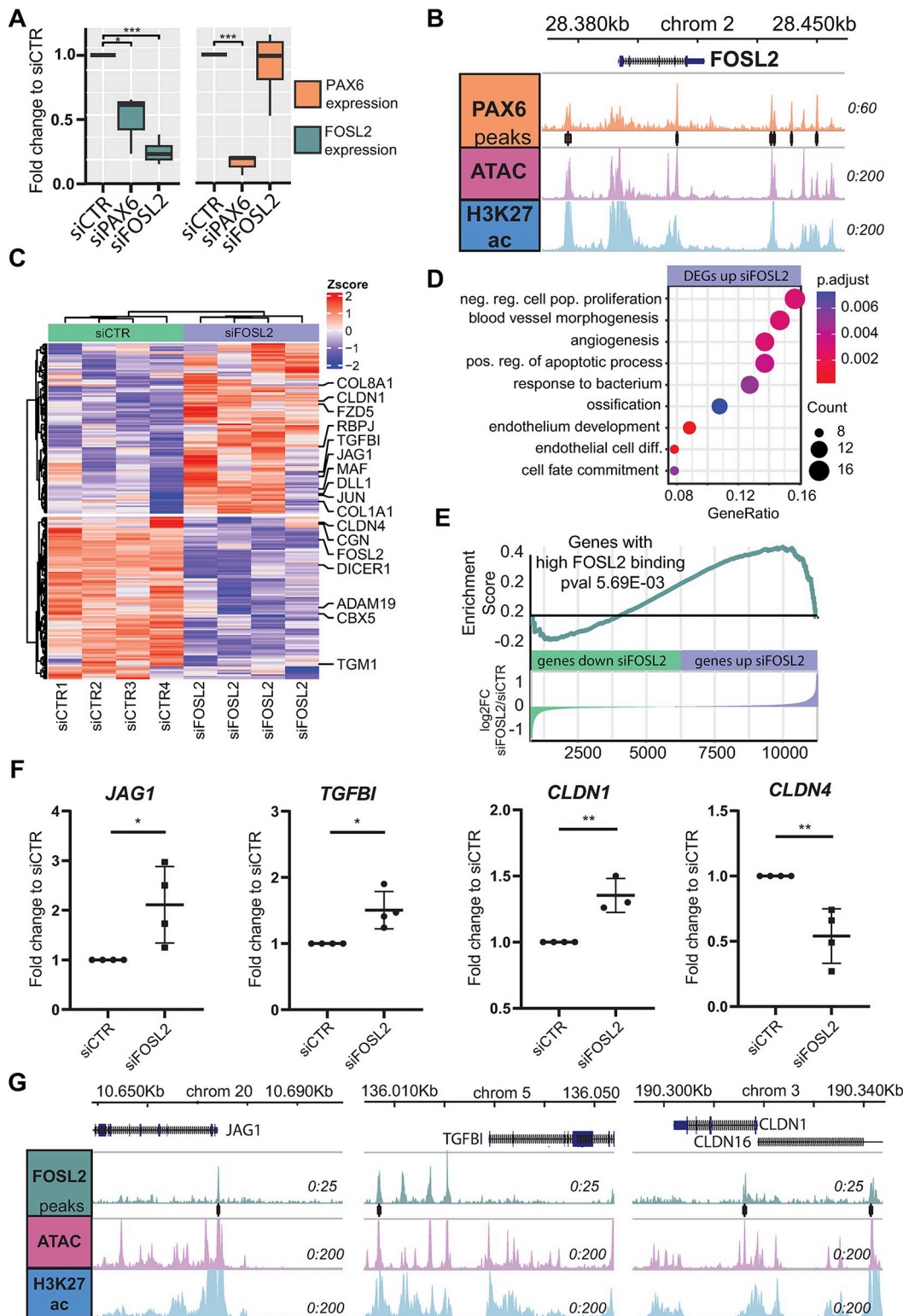

**Fig 6. FOSL2 siRNA knockdown results in deregulation of angiogenesis and tight junction genes.** (A) Normalized transcriptome difference of FOSL2 and PAX6 upon knockdown of FOSL2 (siFOSL2) and PAX6 (siPAX6) (* pval<0.05, ** pval<0.01, *** pval<0.001, paired DESEQ2 differential expression testing). (B) FOSL2 TSS locus with PAX6 binding signal, ATAC seq, and H3K27ac ChIP-seq in LSCs. (C) Heatmap of differentially expressed genes in siFOSL2, with Zscore plotted. (D) GO term enrichment of genes up-regulated in siFOSL2 (E) GSEA enrichment analysis based on the expression

foldchange (siFOSL2/siCTR) of genes with nearby FOSL2 binding signal. (F) Expression of JAG1, TGFBI, CLDN1, and CLDN4 were measured in control LSCs (siCTL) and FOSL2 siRNA-knock down (siFOSL2) samples ($n$ = 4). Values represent the fold change difference of siFOSL2/siCTR and were normalized to internal housekeepers GAPDH and ACTB (* pval<0.05, ** pval<0.01, *** pval<0.001, unpaired $t$ test analysis). The data underlying this figure can be found at S10 Table. (G) JAG1, TGFBI, and CLDN1 TSS loci with FOSL2 CUT&RUN signal, ATAC-seq, and H3K27ac ChIP-seq in LSCs. For the underlying data see S10 Table, GEO GSE242990, GSE236440, and the Zenodo entry [51]. CUT&RUN, cleavage under targets and release using nuclease; GO, Gene Ontology; GSEA, gene set enrichment analysis; LSC, limbal stem cell; TSS, transcription start site.

RUNX1 and SMAD3 [32]. Although SMAD3 is also expressed in KCs, it has higher expression in LSCs, and therefore, it is identified to have a higher influence score LSCs than in KCs in our study. OTX1 is an important TF for regulating the neural lineage [61,62]. In mice, both Otx1 and its ortholog Otx2 are vital for tissue specification during eye development, particularly of the retinal pigmented epithelium [63,64]. ELF3 has previously been linked to *KRT12* and *KRT3* regulation [65] and is one of the TFs identified to play a major role in LSC stratification [39].

FOSL1, along with FOSL2, a novel candidate for corneal disease in our study, and other TFs from the FOS and JUN families, are known to be important in epidermal cells. These TFs can form the AP1 complex together with JUN factors [66], which regulates various biological processes, including epidermal stratification [67]. In our differential motif analysis, the FOS motif that can be bound by FOS, FOSL1, and FOSL2, is the most abundant motif enriched in LSCs, as compared to KCs, highlighting the potential role of FOS, FOSL1, and FOSL2 in LSCs.

We found that only a small number of TFs have been identified as KC specific, including GATA3, CEBPA, and HOX genes. Among these TFs, GATA3 and CEBPA have been reported to have a role in the epidermis. CEBPA has been shown to regulate p63 expression [68], while GATA3 is regulated by p63 in the epidermis [69–71]. Interestingly, GATA3 was up-regulated in aniridia patient LSCs, which is in line with the concept that loss of PAX6 in aniridia LSCs could lead to a KC-like signature. However, our GSEA analyses did not show significant gene expression similarity between up-regulated genes in aniridia LSCs and KCs, indicating that aniridia patient LSCs do not acquire a complete KC fate. Surprisingly, we detected HOXA9 as one of the KC-specific TFs. HOX genes are well known in antero-posterior body patterning and segmentation where mesodermal genes and cells are mainly involved [72,73], but little is known about their function in the epidermis. One plausible interpretation is that, as LSCs are from the eye and KCs are from the trunk of the body, the detection of HOX9 for the KC fate simply marks the positional information along antero-posterior axis. Nevertheless, the lower number of KC-specific TFs, as compared to LSC-specific TFs, indicates that repression of LSC-specific TF expression is critical for the KC fate. This is in line with our observation that LSC-specific TFs such as the *PAX6* locus is completely covered by H3K27me3, probably via polycomb repression. It is also worth noting that the ANANSE prediction tool used in this study is unable to reliably predict TFs with transcriptional repression functions [53], which limits the identification of TFs to repress LSC genes, if there are any in KCs. Consistent with this, ANANSE did not detect repressive TFs such as OVOL2 [74] and SOX9 [75] that have been shown to play roles in LSCs.

We anticipated that TFs that are shared but important to both LSCs and KCs could be not identified through pairwise gene regulatory network comparison between LSCs and KCs. For detecting these shared TFs, we compared both cell types to pluripotent stem cells in the gene regulatory network analysis. This approach indeed resulted in a significant number of shared epithelial TFs including FOSL2, p63, EHF, TFAP2A, KLF4/5, FOS, JUN, RUNX1, and FOXC1. Many of these have previously been linked to important functions in both epidermis and cornea [12–14,16,32,76–80].

The key role of TFs in cell fate control is often demonstrated by their association with developmental diseases. PAX6, FOXC1, and p63 are known to be associated with corneal opacity [20,33,37], and FOSL2 is a novel corneal opacity gene identified in this study. Except for PAX6, both FOXC1 and FOSL2 are shared between LSCs and KCs. FOSL2, PAX6, and FOXC1 seem to regulate most identified disease genes. For PAX6 and FOXC1, this is consistent with their broad expression patterns in the eye and the phenotypic heterogeneity and overlap linked to *FOXC1* and *PAX6* mutations, e.g., iris and corneal defects and higher prevalence of glaucoma [33,37], reinforcing a common regulatory network shared by the 2 TFs [81]. As for PAX6- and p63-associated disorders, although *PAX6* and *TP63* mutations are known to cause corneal opacity, other phenotypes are quite distinct, fully in line with their gene expression in different tissues, e.g., *PAX6* in the cornea epithelium, iris, retina, pancreas, and parts of the central nervous system, and *TP63* in the cornea epithelium, skin epidermis, and other stratified epithelia [20,33]. Although FOXC1 was annotated as a shared TF of LSCs and KCs in our study, its relevance for the skin and cornea might still be different. In our analysis, FOXC1 had a higher influence score in LSCs, probably due to its higher expression in LSCs. This is in line with skin phenotypes not being reported in anterior segment dysgenesis associated with *FOXC1* mutations.

In our analysis, we identified 2 TFs that are part of the AP1 complex, FOSL1 and FOSL2. We identified FOSL2 as a shared epithelial and novel TF associated with corneal opacity, which was corroborated by in vitro and in vivo protein expression. Furthermore, we identified FOSL2 as a direct target gene of PAX6, and it was down-regulated in PAX6-deficient cells, enforcing the role of FOSL2 in corneal pathology. Interestingly, in our analyses to search disease-associated target genes of key TFs, FOSL2 was predicted to have a potential influence in other eye diseases, similar to the prediction of PAX6 in our study. Like PAX6 that can influence multiple tissues of the eye [29], FOSL2 may be a candidate gene for other eye diseases. This is in line with the reported link between keratoconus and down-regulation of FOSL1 [82], an AP1 complex partner of FOSL2 [83].

In mice, Fosl2 has been linked to extensive dermatosis [83] and abnormalities in the cornea and anterior segment, but these were mainly overexpression studies [84], while in humans *FOSL2* has been linked to skin, with a recent GWAS study linking SNP-associated with increased FOSL2 expression to a higher likelihood of eczema [85]. More importantly, *FOSL2* truncating variants have recently been linked to a neurodevelopmental syndrome with multiple ectodermal symptoms including scalp aplasia cutis (absence of scalp skin) and tooth and nail abnormalities [86]. Patients also presented with congenital cataracts, but no corneal phenotype was discovered; however, these often manifest at an older age than the patients' age reported in the study.

In this study, we have found a predicted-pathogenic heterozygous missense variant in *FOSL2* in a proband with corneal opacity. Upon siRNA-mediated knockdown in LSCs, we observed that reduced FOSL2 led to an up-regulation of pro-angiogenesis genes like *JAG1* and *TGFBI*. Furthermore, FOSL2-binding sites were also found in their associated loci, indicating a role of FOSL2 in repressing angiogenesis. Corneal neovascularization is one of the hallmarks of LSCD and corneal opacities. FOSL2 has been identified to promote VEGF-independent angiogenesis in fibroblasts [87], and *FOSL2* knock down in glioma endothelial cells increased blood tumor barrier permeability [88]. The latter study also showed that FOSL2 binds to the promoters and down-regulates the expression of several tight junction genes *TJP1*, *OCLN*, and *CLDN5*, increasing barrier permeability. In accordance, we also detected altered expression of many tight junction and barrier genes in our siFOSL2 LSCs, such as *CLDN1*, *CLDN4*, or *TGM1*. Tight junctions are essential for the cell–cell contact and permeability barrier function in epithelial cells, and defects in tight junctions are linked to several epithelial diseases [89,90].

Therefore, deregulation of tight junction genes and up-regulation of pro-angiogenic genes upon *FOSL2* knockdown strongly support FOSL2 relevance in LSCs identity and the corneal opacity phenotype, as well as the skin phenotype in reported patients [86]. The role of FOSL2 in the skin epithelial phenotype would require further investigation.

In this study, we identified shared and cell type-specific epithelial TFs that are important in determining cell fate of LSCs and KCs. Furthermore, we provided evidence for potential TFs and regulatory mechanisms in corneal diseases by uncovering a novel role for FOSL2 in corneal opacity. We therefore show proof of principle that these key LSC TFs and their gene regulatory networks can be leveraged as a resource to unveil pathomechanisms behind corneal opacity diseases.

## Materials and methods

### Ethical statement

All procedures for establishing and maintaining human primary keratinocytes were approved by the ethical committee of the Radboud University Medical Center ("Commissie Mensgebonden Onderzoek Arnhem-Nijmegen") (CMO-nr:2004/132). All donors have given informed consent in written forms.

All procedures for establishing and maintaining LSCs were conducted according to the principles expressed in the Declaration of Helsinki. The use of LSCs extracted from corneal scleral donor rims was approved by the Ethics Committee of the Saarland (Number 226/15 and 110/17). Aniridia patients consented in written form to research limbal biopsy during ocular surgery.

Permission to use the donor corneas for research purposes was given by the donor's next of kin by signing a written consent form released by the Regional Transplant Service and the National Transplant Service, without ethics committee approvals. According to Article 4 of the Italian Law 91 of 1st April 1999, donor corneas that cannot be used for transplantation (i.e., unsuitable for biological reasons or anamnestic reasons) can be used for research purposes if the aim is to ameliorate corneal transplantation or progress towards a cure of corneal diseases. The human corneas used in this study were unsuitable for transplantation and obtained by Fondazione Banca degli Occhi del Veneto (www.fbov.org; Venice, Italy) for research purposes.

### KC and LSC cell culture in vitro

KCs were isolated and cultured as previously described [16]. Briefly, after isolation, primary KCs were cultured in Keratinocyte Basal Medium supplemented with 100 U/ml Penicillin/Streptomycin, 0.1 mM ethanolamine, 0.1 mM O-phosphoethanolamine, 0.4% (vol/vol) bovine pituitary extract, 0.5 μg/ml hydrocortisone, 5 μg/ml insulin, and 10 ng/ml epidermal growth factor. Medium was refreshed every other day until the cells were 90% confluent.

Limbal tissues were acquired as previously described [91]. Two aniridia limbal tissue single biopsies were obtained from the superior limbus during penetrating keratoplasty from 2 patients with congenital aniridia as previously described [92]. Genetics of the aniridia patients were identified to be c.33delC p.Gly12Valfs*19 (NM_000280.2) for AN55 and c.990_993dup p. Met332Alafs*10 for AN40 (S1 Table). Both patients their corneas were in Lagali AAK-Stage 4 during the Keratoplasty [31].

Cell isolation was performed as previously described [91]. Briefly, limbal tissue was digested in collagenase A solution (4 mg/ml) in KSFM (Thermo Fisher Scientific; Waltham, Massachusetts, United States of America) for 20 h at 37˚C. Cell suspensions were filtered through a use of Flowmi micro strainer (SP Bel-Art; Wayne, New Jersey, USA). LSC clusters were dissociated with trypsin-EDTA (0.05%) solution and cultivated in KSFM. Medium was refreshed every

other day. Subconfluent (80% to 90%) limbal epithelial cells were harvested at passage 3. Aniridia LSCs were harvested at passage 3.

Next to this approach, other LSC samples (LSC-Aberdam, S1 Table) were isolated from postmortem donated peripheral corneal epithelium and cultured as previously described [50]. Briefly after isolation, they were expanded and cultured in KSFM (Gibco, Life Technologies) supplemented with 25 μg/ml bovine pituitary extract (BPE; Gibco, Life Technologies), 0.2 ng/ml epidermal growth factor (EGF, Peprotech, Neuilly-sur-Seine, France), 0.4 mM CaCl$_2$, 2 mM Glutamine (Gibco, Life Technologies), and 100 U/ml Penicillin/Streptomicin (Gibco, Life Technologies). Medium was refreshed every other day until the cells were 90% confluent.

Finally for CUT&RUN of LSCs, LSCs were isolated and cultured as described previously [93,94]. Briefly, corneal limbal tissue was dissected from human donor corneas, cut into small pieces, and incubated with 10 ml trypsin (0.05% trypsin—0.01% EDTA solution, Life Technologies, 25300–062) for 30 min at 37˚C for 4 consecutive times. Every time, the supernatants containing the cells were collected, neutralized, centrifuged at 1,000 rpm for 5 min and cells co-cultured with a feeder layer (consisting of lethally irradiated 3T3-J2 cells) in LSC growth medium. The LSC growth medium contains 2:1 Dulbecco's Modified Eagle Medium (Life Technologies, 21969035) and F12 (Life Technologies, 21765029), 10% Fetal Bovine Serum (Life Technologies, 10099–141), 50 mg/ml penicillin-streptomycin (Life Technologies, 15140122), 4 mM glutamine (Life Technologies, 25030081), 0.18 mM adenine grade I (Pharma Waldhof GMBH, 4010-21-2), 0.4 mg/ml hydrocortisone (Flebocortid Richter, Sanofi, AIC013986029), insulin (Humulin R, Lilly, Canada, HI0210), 2 nM triiodothyronine (Liotir, IBSA, AIC036906016), and 8.1 mg/ml cholera toxin QD (List Biological Laboratories, 9100B). Cultures were incubated at 37˚C with 5% CO$_2$. Medium was changed every 2 days with further addition of 10 ng/ml EGF (Cell Genix GmbH, Germany 1416–050). Cultures were switched to KSFM and used after 3 passages.

## siRNA treatment

Primary human limbal epithelial stem cells were isolated from healthy donors and cultivated in 6-well plates in KSFM medium (Cat. Nr. 17005042, Thermo Fisher, Gibco, Life Technologies, Paisley, United Kingdom) supplemented with 500 ng EGF, 12.5 mg BPE (Cat. Nr. 37000015, Thermo Fisher, Gibco, Life Technologies, Paisley, UK) and 0.1% P/S. After reaching 70% to 80% confluence, cells were transfected using Lipofectamin 2000 transfection reagent. For this purpose, 5 μl Lipofectamin was added to 250 μl Optimem and incubated for 5 min, at room temperature (RT). In a separate tube, siRNA of interest (information and concentrations used are summarized in Table 1) was diluted in 250 μl Optimem. The Lipofectamin was added to the siRNA mixture and incubated for 20 min, RT. After the incubation time, transfection mixture was added dropwise to the cells. The cells were incubated at 37˚C and 5% CO2 and medium were changed after 24 h. Cells were collected 48 h after transfection for further analysis.

**Table 1. siRNA oligonucleotide sequences for FOSL2 and PAX6 knockdown.**

|  | siRNA | siRNA sequence | Company |
| --- | --- | --- | --- |
| FOSL2 siRNA | 100 pmol | Sense: CGAACCUCGUCUUCACCUAtt<br>Antisense: UAGGUGAAGACGAGGUUCGag | Ambion |
| Ctrl siRNA for FOSL2 | 100 pmol | Catalog No. 4404021 | Ambion |
| PAX6 siRNA | 15 pmol | combination of 2 different siRNAs: 5′CCUGGCUAGCGAAAAGCAAUU<br>5′UGGGCGGAGUUAUGAUACCUU | MWG Eurofins |
| Ctrl siRNA for PAX6 | 15 pmol | 5′AGGUAGUGUAAUCGCCUUGUU | MWG Eurofins |

## Bulk RNA-seq

Total RNA was isolated using the Quick–RNA MicroPrep kit (Zymo Research), according to the manufacturer's protocol. RNA concentrations were measured using the the DeNovix DS-11FX spectrometer, and 500 ng of RNA was prepared for sequencing using the KAPA RNA HyperPrep Kit with RiboErase (Kapa Biosystems). Libraries were sequenced on the NextSeq 500 (Illumina), generating an average of 15 to 20 million reads per sample.

## Single-cell RNA-seq library preparation and sequencing

A single-cell suspension was made using trypsin, after which cells were filtered using a 40 µm filter to remove cell clumps. Cells were stained with 7-AAD. The live cells were selected for and FACS-sorted onto 384-well plates containing primers with unique molecular identifiers, according to the SORT-Seq protocol [95]. Plates were spun down (1,200 × g, 1 min, 4˚C) and ERCC spike-in mix (1:50,000) was dispensed by a Nanodrop (BioNex) into each well, and 150 nl of the Reverse Transcription (RT) mix was dispensed into each well. Thermal cycling conditions were set at 4˚C 5 min; 25˚C 10 min; 42˚C 1 h; 70˚C 10 min. The library of each plate was pooled together and the cDNA was purified using AmpureXP (New England BioLabs) beads. Overnight in vitro transcription (Ambion MEGA-Script) was carried out at 16˚C, with the lid set at 70˚C. An exonuclease digestion step was performed thereafter for 20 min at 37˚C, followed by fragmentation of the RNA samples. After a beads cleanup, the samples were subjected to library RT and amplification to tag the RNA molecules with specific and unique sample indexes (Illumina), followed by a final beads cleanup (1:0.8, reaction mix: beads) and the sample cDNA libraries were eluted with DNAse free water. Libraries were quantified using the KAPPA quantification kit following manufacturer's protocol after which the plates were sequenced on the NextSeq 500 (Illumina) for 25 million reads per plate.

## ChIP-seq

Chromatin for ChIP was prepared as previously described [96] with minor modifications. On average, 0.5 M cells were used in each ChIP. Antibodies against H3K27ac (Diagenode #C15410174, 1.2 µg), H3K4me3 (Diagenode #C15410003, 1 µg), H3K27me3 (Diagenode #C15410069, 1.5 µg), p63 (Santa Cruz #H129, 1 µg, recognizing the C-terminal α tail of p63) were used in ChIP assay. Afterwards, 5 ng DNA fragments were pooled and proceeded on with library construction using KAPA Hyper Prep Kit (Kapa Biosystems #KK8504) according to the standard protocol. The prepared libraries were then sequenced using the NextSeq 500 (Illumina) according to standard Illumina protocols.

## Immunofluorescence stainings

Human donor corneas were fixed with 4% paraformaldehyde overnight at 4˚C, soaked in increasing gradients of sucrose solutions (7.5%, 15%, and 30% in PBS, at least 30 min each at 4˚C, the last one overnight) and then embedded in OCT compound, frozen, and cut into 5- to 7-mm sections. Sections were permeabilized in a solution containing 0.5% Triton X-100 for 10 min, incubated in 5% BSA (Sigma-Aldrich A7906) for 1 h at RT and then incubated overnight at 4˚C with primary antibodies. The following primary antibodies (all diluted at 1:100) were tested: PAX6 (rabbit, Biolegend Poly19013); FOSL1 aka Fra-1 (C-12) (mouse, Santa Cruz Biotechnology, sc-28310); Anti-FOSL2 (rabbit, Sigma, HPA004817); p63 (mouse, DAKO, M7317). Sections were extensively washed in 1× PBS and incubated with secondary antibodies (Alexa Fluor 488 anti-rabbit A11008, Rhodamine Red-X goat anti-mouse IgG R6393, Rhodamine Red-X goat anti-rabbit IgG R6394, Alexa Fluor 488 goat anti-mouse A11001, all from

Invitrogen, all diluted at 1:500) for 45 min at RT. Sections were eventually mounted with medium containing DAPI Fluoromont-G (EMS, Società Italiana Chimici, Rome, Italy, 17984–24) and pictures taken and evaluated through an Eclipse Ti Nikon microscope (Nikon, Amstelveen, the Netherlands).

LSCs and primary keratinocytes were grown in cell culture chamber slides (Corning, New York, USA), and, when confluent, fixed using 4% PFA solution. Immunostainings were performed as previously described [97]. Briefly, cells were washed, permeabilized for 15 min with 0.1% Triton X-100, washed again, and blocked for 2 h in 1% FBS. Primary antibody incubation was done overnight at 4˚C with the following antibodies diluted in blocking buffer: PAX6 (1:300, 901301 Biolegend), P63 (1:100, sc8344 SCBT), FOSL1 (1:100, sc20310, SCBT), and FOSL2 (1:200, HPA004817 Sigma Aldrich). Cells were washed thoroughly and incubated with secondary antibodies AlexaFluor 488 (1:400, A11008 Invitrogen) and AlexaFluor 647 (1:400, A31571 Invitrogen, California, USA). Slides were mounted with Vectashield (VectorLabs, California, USA) and analyzed with Zeiss AI Sample Finder (Zeiss, Jena, Germany). Fiji/ImageJ software (National Institutes of Health, Maryland, USA) was used for overlay images preparation.

## CUT&RUN

CUT&RUN was performed using the CUT&RUN Assay Kit (Cell Signaling Technology #86652) and according to the manufacturer instructions. Briefly, 200,000 live primary LSCs were used per immunoprecipitation (IP). For FOSL2 IPs, 10 ml of antibody (D2F1E, Rabbit mAb #19967) was used per IPs, 2 IPs were performed per replica to ensure a sufficient amount of DNA. The negative control IgG included in the kit was used to check for enrichment by quantitative PCR.

For high-throughput sequencing, libraries were prepared using KAPA HyperPrep Kit (Kapa Biosystems #KK8504) using 5 ng of DNA. Libraries concentration and fragment repartition were checked by automated electrophoresis (Bioanalyzer Instrument, Agilent). Libraries were sequenced using the NextSeq 500 (Illumina) and according to their protocol.

## Reverse transcription quantitative PCR (RT-qPCR) validation

Approximately 250 ng RNA were used for cDNA synthesis using the iScript cDNA synthesis kit (BioRad, California, USA). Quantitative PCR (qPCR) was carried out using 4 μl cDNA (diluted 1:20) with iQ SYBR Green Supermix (BioRad, California, USA) and specific primers (Biolegio, Nijmegen, NL):

*GAPDH* Forward (Fw): AAG GAG TAA GAC CCC TGG ACC A
*GAPDH* Reverse (Rv): GCA ACT GTG AGG AGG GGA GATT
*ACTB* Fw: TTC TAC AAT GAG CTG CGT G
*ACTB* Rv: GGG GTG TTG AAG GTC TCA AA
*JAG1* Fw: GAT CGC CTG CTC AAA GGT CT
*JAG1* Rv: GAC TGG AAG ACC GAC ACT CG
*TGFBI* Fw: CCT TTG AGA CCC TTC GGG CTG
*TGFBI* Rv: TCG AAG GCC TCA TTG GTC GG
*CLDN1* Fw: CCC CAG TCA ATG CCA GGT ACG
*CLDN1* Rv: TCG GGG ACA GGA ACA GCA AA
*OCLN* Fw: TCT AGG ACG CAG CAG ATT GGT
*OCLN* Rv: TCA GGC CTG TAA GGA GGT GG

All disease gene qPCRs were performed using a BioRad CFX96 system following standard cycling conditions. Data were analyzed by the ΔΔCT method, normalized to housekeepers *GAPDH* and *ACTB*.

## FOSL2 and PAX6 siRNA RT-qPCR

RNA extraction were performed using RNA isolation kit (RNA Purification Plus Micro Kit, Cat. Nr. 47700, (Norgen Biotek CORP. Canada)) according to the manufacturer's instructions. Optional DNA column digestion was performed. The eluted RNA samples were stored at −80˚C until further use. NEB One Taq RT-PCR kit (One Taq RT-PCR Kit, New England Bio-labs, Frankfurt, Germany) was used for cDNA synthesis, following the manufacturer's instructions. Samples were stored at −20˚C. The qPCR mix contained 1 μl of the specific primer solution, 5 μl SYBR Green Mix (Qiagen N.V., Venlo, the Netherlands), and 3 μl nuclease-free water (total volume: 9 μl). To investigate the mRNA expression levels, 1 μl cDNA of interest was added to the qPCR mix. The "QuantStudio 5 real-time PCR system" (Thermo Fisher Scientific, Waltham, Massachusetts, USA) was used for the qPCR reactions. The amplification conditions were 95˚C for 10 s, 60˚C for 30 s, and 95˚C for 15 s (40 cycles). Samples were measured in duplicates and experiments were repeated for at least 5 donors. Values were normalized to the expression value of TATA-Box binding protein as an endogenous control gene using the ΔΔCT method and the fold change (2ΔΔCT-value) was used for statistical analysis. All primers were purchased from Quantitect Primer assay (Qiagen N.V., Venlo, the Netherlands; PAX6: QT00071169, FOSL2: QT01000881, and TBP: QT00000721).

## RNA-seq, ATAC-seq ChIP-seq, and CUT&RUN data preprocessing

Preprocessing of reads was done automatically with workflow tool seq2science v0.7.1 [98]. Paired-end reads were trimmed with fastp v0.20.1 [99] with default options. Genome assembly GRCh38.p13 was downloaded with genomepy 0.11.1 [100]. Public samples were downloaded from the Sequence Read Archive [101] with help of the NCBI e-utilities and pysradb [102]. The effective genome size was estimated per sample by khmer v2.0 [103] by calculating the number of unique kmers with k being the average read length per sample. scATAC fastq files were merged to pseudobulk by combining all fastq files from each plate using the bash command cat.

Reads of ChIP-seq, CUT&RUN, and ATACseq were aligned with bwa-mem v0.7.17 [104] with options '-M'.

Reads of RNAseq samples were aligned with [105] with default options. Afterwards, duplicate reads were marked with Picard MarkDuplicates v2.23.8 [106]. General alignment statistics were collected by samtools stats v1.14 [107]. Mapped reads were removed if they did not have a minimum mapping quality of 30, were a (secondary) multimapper or aligned inside the ENCODE blacklist [108]. RNAseq sample counting and summarizing to gene level was performed on filtered bam using [109]. Sample sequencing strandness was inferred using [110] in order to improve quantification accuracy.

ATAC samples were tn5 bias shifted by seq2science. ChIP, CUT&RUN, and ATAC sample peaks were called with macs2 v2.2.7 [111] with options '—shift -100—extsize 200—nomodel—keep-dup 1—buffer-size 10000' in BAM mode. The effective genome size was estimated by taking the number of unique kmers in the assembly of the same length as the average read length for each sample. Narrowpeak files of ChiP-seq and CUT&RUN biological replicates belonging to the same condition were merged with the irreproducible discovery rate v2.0.4.2 [112]. ATAC-seq samples were correlated seq2science its DESeq2 reads per peak spearman correlation clustering (S5F Fig).

## Single-cell RNA-seq data preprocessing

Single-cell libraries were preprocessed using the cellseq2 pipeline. Briefly, reads were aligned using star to the GRCh38.p13 genome. After which cells were quality controlled using Seurat, filtering cells on ERCC reads, genes measured and transcripts per cell. After visualization of the lack of heterogeneity by Umap, pseudobulk count data was generated by summing all the cells their UMI counts. Cellular heterogeneity was assessed using the analysis file Generate_scRNAseq_pseudobulk.Rmd. Finally, single-cell and bulk gene count tables were merged for a combined bulk and pseudobulk analysis.

## RNA-seq data analysis and normalization

The bulk and pseudobulk count tables were merged on gene names, keeping all gene names detected. Due to potential sex differences between donors, genes located on chromosome X and Y were removed. Finally, genes with less than 10 counts per row were removed. Variance visualization was performed using sample distance and PCA. For quality control, sample variance and distance were visualized before and after removing technical variance due to different sequencing methods. Limma [113] was used to remove these batch effects.

Rld normalization was used for normalizing gene intensities. Between all conditions differentially expressed genes were detected using Deseq2 [114]. Non-batch corrected count tables were used for identifying the differentially regulated genes. Ashr log2 fold change shrinkage [115] was used to shrink the Log2 fold change values. Differentially regulated gene detection cutoffs were set as an adjusted *p*-value of 0.01 or lower, and an absolute log2 FC of 0.58 and larger.

The packages complex Heatmaps [116] and circlize [117] were used to visualize the differentially expressed genes. Subsequently, Progeny enrichment [47] was performed to quantify signaling pathway target gene enrichment. Clusterprofiler [44] was run for GO term enrichment on differentially expressed genes of each comparison. Finally, foldchange of all genes were used to generate a gene list for GSEA enrichment of the MSigDB collections [46]. Gene names were mapped to ENTREZID using AnnotationDbi [118] and these were used to run KEGG pathway enrichment. The enriched pathways were visualized using pathview [119].

## Identification of CREs

In order to identify CREs, ATAC-seq was used. Bulk and scATAC data were merged from in vitro expanded KCs and LSCs. Next to the generated datasets, publicly available data were incorporated. To prevent a sequencing depth bias, the top 100.000 ATAC peaks from each cell type were combined, the overlapping peak summits were merged, and histone modifications in varying window sizes around these ATAC peaks were quantified using histone ChIP-seq datasets (Fig 2A). For ATAC signal quantification, the ATAC intensity was quantified in 200 bp around the peak summits, and for the promoter mark H3K4me3 and the enhancer mark H3K27ac, a 2kb window was used and finally for the repressive H3K27me3 mark, a 5 kb window was used for quantification (Fig 2A). This resulted in an extensive dataset containing CREs and their respective histone modification signal intensity.

Differential CREs were identified for the ATAC-seq and H3K27ac reads by running DESEQ2 on the read counts within the defined windows and identified regions (adjusted *p*-value <0.05). For H3K4me3 and H3K27me3 signals, differential CREs were identified with 2 steps. First, the histone mark distribution was plotted, and CREs with a low to no histone signal were disregarded. Next, high activity regions with variable signal were selected (S5 Fig).

Variable CREs were linked to genes with 2 approaches:

1. CREs were linked to all the TSS regions within a 100 kb window using bedtool window [120] after which the CREs per TF were distance weighted and summed based on the ANANSE distance weighing approach including promoter peaks [53].

2. CREs were linked to the closest TSS region within 20 kb using bedtool closest [120].

After linking the regions to genes in both approaches, intensity scores were printed to a CSV file. Heatmaps and GO term enrichments were generated in R using clusterprofiler and complex heatmap.

### Single-cell ATAC-seq

A single-cell suspension was made using trypsin. After which cells were filtered using a 40 μm filter to remove cell clumps. The protocol on from Chen and colleagues [121] was used to sequence single-cell ATAC. Briefly, 50.000 cells were tagmented in bulk in 45 μl of tagmentation mix (20 mM Tris (pH 7.6), 10 mM magnesium chloride 20% Dimethylformamide), 5 μl of tagmentation protein and 0.25 μl of Digitonin. Cells were tagmented for 30 min at 37°C and 800 rpm. Tagmentation was stopped by adding 50 μl of tagmentation stop buffer (10 mM Tris-HCL (pH 7.8) and 20 mM EDTA). Cells were stained with DAPI and DAPI positive cells were FACs sorted in 384-well plates containing Nextera primers with unique molecular identifiers, NACL ProteinaseK and SDS page. Plates were spun down ($1200 \times g$, 1 min, 4°C) and were incubated for 15 min at 65°C; 4 μl of Tween20 was added, and 2 μl of H20 was added, and finally 10 μl of NEBNext High-Fidelity 2X PCR Master Mix was added to each well. Thermal cycling conditions were set at 72°C for 5 min, 98°C for 5 min, and then, 20 repeats of 98°C for 10 s, 63°C for 30 s, and 72°C for 20 s.

Plate libraries were pooled and purified using a Qiagen PCR purification kit with adjusted buffer volumes according to Chen and colleagues [121]. After the column cleanup, a final beads cleanup was performed using AmpureXP (New England BioLabs) beads and the sample cDNA libraries were eluted with DNAse free water. Libraries were quantified using the KAPPA quantification kit following manufacturer's protocol after which the plates were sequenced on the NextSeq 500 (Illumina) for 30 million reads per plate.

### Motif analysis

The Gimme motifs database was pre-filtered [52] to include only motifs linked to TFs which were expressed in either KC and/or LSCs (using a cutoff of at least 10 counts in total). When multiple motifs mapped to a TF, the most variable motif was used. In case multiple TFs mapped to a motif, the most differential TF on the transcriptome was annotated to the motif. All highly variable CREs their log10 quantile normalized values were used as an input for Gimme maelstrom motif enrichment analysis.

### ANANSE analysis

For the gene regulatory network analysis, all the called ATAC peaks were used, merging summits and excluding peaks on the chromosomes GL, Un, KI, MT, X, and Y due to potential donor sex differences. Next, ANANSE binding was ran using all the peaks as potential enhancer regions and using both ATAC and H3K27ac signals to predict potential TF binding. To select the TF binding model, a Jaccard similarity score of min 0.2 was used, to minimize the false-positive models used to predict TF binding. For ANANSE network, the ANANSE binding files were combined with the RNAseq TPM files. This included all bulk RNAseq samples of KCs, LSCs, and ESCs (S1 Table). In the case of the in vivo pseudobulk data, FPKM values were used based on the UMI tables.

Finally, ANANSE influence was ran using the top 500.000 differential edges between networks. Deseq2 was ran on the countfiles of each comparison to identify differential genes needed for ANANSE influence. To prevent missing values, for the final ESC-KC, ESC-LSC, KC-LSC, and LSC-KC comparison, all differential edges were taken from each comparison and used to reran each comparison with all these edges included. This prevented missing values in the differential networks while comparing different differential networks.

TF hierarchy was estimated using the TF-target TF binding score generated by the influence command running the–full-output flag. This represents the motif, ATAC&H3K27ac signal intensity in the target TF locus and is excluding the difference in expression. The Delta binding score was calculated by subtracting the score of a TF-gene interaction within 1 GRN with the score of the interaction within the other GRN.

The delta binding score of the ESC-KC and ESC-LSC comparisons were averaged. If this average was higher than the delta binding score of KC-LSC and LSC-KC, the interaction was classified as "shared epithelial," if the delta binding score was highest in LSC-KC, it was classified as "KC specific," if the delta binding score was highest in KC-LSC, it was classified as "LSC specific."

## Single-cell RNA-seq analysis of the epidermis and the cornea

The raw sequencing data was downloaded from GEO and split it into fastq files using seq2science. Cellranger count was run with Cellranger 6.0.1 to retrieve the matrix, barcodes, and features files necessary for Seurat [122] analysis in R. scRNA-seq cells were selected with a minimum count of 2,000, a feature number above 1,000, and a mitochondrial percentage below 30 percent. Cell cycle scoring was performed using Seurat CellCycleScoring() feature with the cell cycle genes from Tirosh and colleagues [123]. All cells not in the G1 phase were removed. Leiden Clustering was performed and cell clusters were annotated based on described marker genes.

For the data of the epidermis, cell clusters were selected with high KRT14, KRT5, and low KRT1 and KRT10 expression as basal KCs. From the cornea dataset, cell clusters with high S100A2 with PAX6 and TP63 expression and without CPVL expression were selected as LSCs. The in vivo versus in vitro fold change difference plot was generated by loading deseq2 result tables to identify the TF fold changes.

## ChIP-seq and CUT&RUN data analysis

ChIP-seq and CUT&RUN peaks were called with MACs2 and validated by IDR (preprocessing). Next for each peak summit reads were counted in 200 bp windows across each summit. Values were log-transformed and quantile-based normalized. Peaks were linked to TSS regions in 100 kb, using bedtools window. Afterwards, they were distance weighted using the ANANSE distance weighing approach. When genes did not have ChIP-seq peaks within a 100 kb window, they got an intensity score of 0.

Disease gene lists were collected from the EyeDiseases database [57], including all disease gene lists of more than 20 genes. A one-sided Mann–Whitney U test was performed to test the hypothesis that the disease genes have more TF binding than the other genes in the genome. Of each significant hit, the top 5 of most bound gene loci were outputted to a list. And the final list was used to generate a dotplot in R (hipseq_intensity_npeak_dotplot.Rmd). Alternatively, ChIP-seq/CUT&RUN peaks were mapped to the gene TSS start site using bedtool closest. Next, disease genes mapped versus non-mapped disease genes were compared to all genes mapped versus non-mapped. Using a Fisher exact test.

## Cornea disease gene list

Curated cornea disease list was firstly compiled by retrieving all known genetic disorders affecting the cornea and respective affected genes from "Ophthalmic Genetic Diseases" [124] and then confirmed using available literature in Pubmed (https://pubmed.ncbi.nlm.nih.gov/) and online eye disease (https://gene.vision/) databases. Diseases were grouped as (1) corneal diseases; (2) systemic (or other ocular) disorders with corneal phenotypes; and (3) diseases with secondary cornea involvement (due to exposure or unclear involvement). Genes associated with multifactorial disease keratoconus were added based on literature search mainly of published genome-wide association (GWAS) and linkage (GWLS) studies [124–126]. Curated gene list is available as S7 Table.

## Variant discovery

Participants of the 100,000 Genomes Project were identified for our analyses who had at least one of the following HPO terms or daughter terms present: corneal opacity (HP:0007957), corneal scarring (HP:0000559), Opacification of the corneal stroma (HP:0007759), central opacification of the cornea (HP:0011493), band keratopathy (HP:0000585), central posterior corneal opacity (HP:0008511), corneal crystals (HP:0000531), generalized opacification of the cornea (HP:0011494), peripheral opacification of the cornea (HP:0008011), punctate opacification of the cornea (HP:0007856), and sclerocornea (HP:0000647). A total of 33 probands were identified who remain genetically unsolved. The whole genome sequence data was interrogated for single-nucleotide variants (SNVs) and indels (insertions or deletions), copy number variants (CNVs), and structural variants as previously described [127]. Filtered variants were annotated using Ensembl Variant Effect Predictor (VEP v99) and prioritized identified variants using scores available from CADD, MutationTaster, Provean, Sift, polyphen2, MetaRNN, DANN, fathmm-MKL. Variant nomenclature was assessed using Variant Validator.

## Supporting information

**S1 Fig. Additional quality control RNA-seq analysis of LSCs and KCs.** (A) PCA plot of RNA-seq samples before batch correction. (B) PCA plot after batch correction. (C) Pearson correlation matrix before batch correction. (D) Pearson correlation matrix after batch correction. (E) Umap dimensionality reduction of scRNA-seq data, visualizing the samples each cell is from on the left, and the cell cycle state on the right. (F) Gene count plot for PAX6, KRT1, KRT19, and TP63 in all KC and LSC samples. (G) TPM gene plots for KRT1, KRT10, in KC and various stratified KC samples ranging from day 2 (KC_strat_1), day 4 (KC_strat_2) and day 7 (KC_strat_3) of stratification, and of KRT12 and KRT3 in LSC and airlifted stratified cornea epithelial cells (CECs). For the underlying data, see S4 Table, GEO GSE206922, GSE206923, and GSE242995.
(PNG)

**S2 Fig. KEGG pathway expression visualizations LSCs and KCs.** (A) TNF signaling pathway component expression FC differences between KC and LSCs. (B) NF-KAPPA B signaling pathway component expression FC differences between KC and LSCs. For the underlying data, see S4 Table, GEO GSE206922, GSE206923, and GSE242995.
(PNG)

**S3 Fig. RNA-seq analysis of LSCs and Aniridia LSCs.** (A) Heatmap of normalized DEG expression between control and aniridia patient LSCs (adjusted pval < 0.05), using k-means clustering with 2 clusters. (B) PCA plot of RNA-seq samples before batch correction. (C) PCA plot after batch correction. For the underlying data, see S4 Table, GEO GSE206922,

GSE206923, and GSE242995.
(PNG)

**S4 Fig. Additional vizualizations regulatory element (CRE) analysis.** (A) Overview of data types used in our analysis. (B) Variable CREs mapped to the closest TSS within 50 kb. Zscore normalized CRE signal intensities and normalized RNA-seq intensities. (C) Heatmap of PROGENy TNF and NF-KB target genes and the Z-score of the quantile normalized histone intensity signal of the closest CRE and the distance weighted enhancer signal. (D) TNF and HOXA9 TSS loci with signals of RNA-seq, ATAC-seq, ChIP-seq of H3K27ac, H3K4me3, and H3K27me3 in KCs and LSCs. For the underlying data, see S5 Table, GEO GSE206918, GSE206920, and the trackhub in the Zenodo entry [51].
(PNG)

**S5 Fig. Additional quality control regulatory element (CRE) analysis.** (A) Quantile normalized intensity score of all ATAC peaks for the varying histone datasets. Including cutoff value for H3K4me3 and H3K27me3. (B) Resulting intensity score for H3K4me3 and H3k27me3 regions. (C) Deseq2 volcano plot of all ATAC & H3K27ac regions. Variance with the variance cutoff for H3K4me3 and H3K27me3. (D) Resulting population of variable regions. (E) Pie chart of region type distribution. For the underlying data, see GEO GSE206918 and GSE206920.
(PNG)

**S6 Fig. TFs Motifs and hierarchy vizualization.** (A) Enriched motifs linked to the various TFs. (B) General epithelial interactions between TFs. Edge Width corresponds with ANANSE binding score predictions. Node color represents RNAseq fold change between LSC and KCs, while node size represents outdegree. (C) Similar to B but with all the LSC-specific interactions. (D) Similar to B but with all the KC-specific interactions. For underlying data, see the Zenodo entry [51].
(PNG)

**S7 Fig. Extra labeled influence plot and outdegree analysis.** (A) ANANSE influence of ESC to KC (x-axis) and ESC to LSC (y-axis), circle size represents a maximum number of target genes in both comparisons. The circle color represents log2FC between LSC/KC. (B) ESC-LSC top TF interaction network generated by ANANSE. (C) ESC-KC top TF interaction network generated by ANANSE. For the underlying data, see the Zenodo entry [51].
(PNG)

**S8 Fig. Additional quality control in vivo single-cell RNA-seq.** (A) Umap of epidermal scRNAseq dataset of Atwood and colleagues. Basal KC cluster used for validation is highlighted. (B) Umap of scCornea atlas of Collin and colleagues, basal LSC cluster used for validation is highlighted. (C) Marker gene expression used to select the basal KCs cluster. (D) Marker gene expression used to select the basal LSC cluster. (E) GO term enrichment of the basal-KC high DEGS enriched vs. the human genome as a background and simplified using simplify. (F) GO term enrichment of the basal-LSCs high DEGS enriched versus the human genome as a background and simplified using simplify. For the underlying data, see GEO GSE155683 and GSE147482 [55,56].
(PNG)

**S9 Fig. FOSL2 variant and protein stainings.** (A) Overview of the FOSL2 transcript and protein, with the location of the variant of unknown significance. (B) FOSL2 and TP63 staining of the central cornea. (C) FOSL1 and FOSL2 staining of the peripheral cornea (D) FOSL1 and FOSL2 staining of the central cornea. (E) FOSL2 and PAX6 staining of the peripheral cornea.

(F) FOSL1 and FOSL2 staining of the central cornea. (G–J) Immunocytochemistry analysis of fixed LSCs and KCs of transcription factors p63 (E), PAX6 (F), FOSL2 (G), and FOSL1 (H). Predicted general TFs p63, FOSL1, and FOSL2 are detected in the nuclei of both cell types while PAX6 is only detected in LSCs. Note that some non-nuclear signal is found in KCs stained with PAX6 but that is considered autofluorescence. DAPI staining (blue) depicts cell nuclei. Scale bar, 100 μm.
(PNG)

**S10 Fig. TF binding disease genes enrichment overview.** (A) Approach for distance weighing and merging of TF ChIP-seqs per TF. This resulted in a TF-disease gene score distribution that was compared to the distribution of all genes with a one-sided Mann–Whitney U test. (B) Approach for linking the ChIP-seq peaks to the closest gene TSS, after which enrichment for disease genes was tested with a Fisher exact test. (C) FDR values of the significant enriched TFs resulting from the ChIP-seq Mann–Whitney U tests and the Fisher exact test. Significant for FDR < 0.1. For the underlying data, see S7 Table, GEO GSE206920, GSE236440, and GSE156272.
(PNG)

**S11 Fig. Additional quality control siRNA treatment and RNA sequencing.** (A) qPCR validation FOSL2 knockdown the data underlying this figure can be found at S10 Table. (B) qPCR validation PAX6 knockdown the data underlying this figure can be found at S10 Table. (C) Pearson correlation matrix siCTR and siPAX6 samples. (D) Pearson correlation matrix siCTR and siFOSL2 samples. (E) PCA plot of RNAseq siPAX6 samples. (F) PCA plot of RNA-seq siFOSL2 samples. (G) Transcripts CLDN7, OCLN, TGM1, and ABCA12 were measured in control LSCs (CTRL) and FOSL2 siRNA-knock down (FOSL2 KD) samples ($n = 4$). Values represent fold change difference of FOSL2 KD to their respective CTRL and were normalized to internal housekeepers GAPDH and ACTB (* pval<0.05, ** pval<0.01, *** pval<0.001, unpaired t-test analysis). For the underlying data, see S10 Table, GEO GSE242990, and GSE236440.
(PNG)

**S1 Table. RNA-seq dataset overview.** Overview of all RNA-seq datasets generated and used, including datatype, origin, medium condition, and GEO accession number.
(DOCX)

**S2 Table. ATAC-seq dataset overview.** Overview of all ATAC-seq datasets generated and used, including datatype, origin, medium condition, and GEO accession number.
(DOCX)

**S3 Table. TF binding datasets ChIP-seq and CUT&RUN overview.** Overview of all TF binding datasets generated and used, including datatype, origin, medium condition, and GEO accession number.
(DOCX)

**S4 Table. Enrichment data in vitro KC vs. LSC comparisons table.** Overview of all in vitro bulk RNA-seq GO term enrichments, GSEA enrichments, and Progeny enrichment scores.
(XLSX)

**S5 Table. Enrichment data CREs in vitro KC vs. LSC comparisons.** Overview of all in CRE GO term enrichments and GSEA enrichment scores.
(XLSX)

**S6 Table. Enrichment data in vivo KC vs. LSC comparisons table.** Overview of all in vivo pseudo bulk RNA-seq GO term enrichments, GSEA enrichments, and Progeny enrichment scores.
(XLSX)

**S7 Table. Curated disease gene list.** Overview of all disease genes identified, including Phenotype MIM number, clinical description, and gene OMIM number.
(XLSX)

**S8 Table. HP terms included and excluded.** Overview of all the HP terms included and excluded, and the number of GEL probands.
(DOCX)

**S9 Table. Variant pathogenicity predictions.** Overview of the FOSL2 variant info and the pathogenic prediction scores.
(DOCX)

**S10 Table. QPCR measurements.** Overview of the qPCR measurements. qPCR data—ratio/fold change compared to each respective LSCs donor.
(XLSX)

## Acknowledgments

We thank M.P.A. Baltissen, L.A. Lamers, and S. Rinzema for operating the Illumina analyzer and initial data demultiplexing, L. Wingens for support with the celseq library preparations, J. Arts, and W.N. Twilhaar for their preprocessing of the single cell RNA-seq objects of the in vivo datasets, J. Niehaus for support in generating the FOSL2 CUT&RUN data.

This research was made possible through access to the data and findings generated by the 100,000 Genomes Project. The 100,000 Genomes Project is managed by Genomics England Limited (a wholly owned company of the Department of Health and Social Care). The 100,000 Genomes Project is funded by the National Institute for Health Research and NHS England. The Wellcome Trust, Cancer Research UK and the Medical Research Council have also funded research infrastructure. The 100,000 Genomes Project uses data provided by patients and collected by the National Health Service as part of their care and support.

## Author Contributions

**Conceptualization:** Jos G. A. Smits, Dulce Lima Cunha, Simon J. van Heeringen, Huiqing Zhou.

**Data curation:** Jos G. A. Smits, Dulce Lima Cunha, Huiqing Zhou.

**Formal analysis:** Jos G. A. Smits, Dulce Lima Cunha, Nicholas Owen, Simon J. van Heeringen.

**Funding acquisition:** Simon J. van Heeringen, Huiqing Zhou.

**Investigation:** Jos G. A. Smits, Dulce Lima Cunha, Maryam Amini, Marina Bertolin, Camille Laberthonnière, Jieqiong Qu, Lorenz Latta, Lauriane N. Roux, Tanja Stachon.

**Methodology:** Jos G. A. Smits, Maryam Amini, Marina Bertolin, Camille Laberthonnière, Jieqiong Qu, Lorenz Latta, Lauriane N. Roux, Tanja Stachon, Simon J. van Heeringen, Huiqing Zhou.

**Project administration:** Simon J. van Heeringen, Huiqing Zhou.

**Resources:** Jos G. A. Smits, Maryam Amini, Marina Bertolin, Camille Laberthonnière, Nicholas Owen, Tanja Stachon, Mariya Moosajee, Nora Szentmary, Simon J. van Heeringen, Huiqing Zhou.

**Software:** Jos G. A. Smits, Simon J. van Heeringen.

**Supervision:** Simon J. van Heeringen, Huiqing Zhou.

**Validation:** Jos G. A. Smits, Dulce Lima Cunha, Maryam Amini, Camille Laberthonnière, Tanja Stachon, Stefano Ferrari.

**Visualization:** Jos G. A. Smits.

**Writing – original draft:** Jos G. A. Smits, Dulce Lima Cunha, Huiqing Zhou.

**Writing – review & editing:** Jos G. A. Smits, Dulce Lima Cunha, Maryam Amini, Marina Bertolin, Camille Laberthonnière, Jieqiong Qu, Nicholas Owen, Lorenz Latta, Berthold Seitz, Lauriane N. Roux, Tanja Stachon, Stefano Ferrari, Mariya Moosajee, Daniel Aberdam, Nora Szentmary, Simon J. van Heeringen, Huiqing Zhou.

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
