## [Editor Report · Decision Letter 0]

20 Mar 2023

Dear Dr Zhou, 

Thank you for submitting your manuscript entitled "Multi-omics analyses identify transcription factor interplay in corneal epithelial fate determination and disease" for consideration as a Research Article by PLOS Biology.

Your manuscript, the accompanying reviews from Review Commons, and your revision plan have all been evaluated by the PLOS Biology editorial staff, as well as by an academic editor with relevant expertise and I am writing to let you know that we are interested in considering a revision of your manuscript. 

However, before we can formally put your manuscript 'under revision', we need you to complete your submission by providing the metadata that is required for full assessment. To this end, please login to Editorial Manager where you will find the paper in the 'Submissions Needing Revisions' folder on your homepage. Please click 'Revise Submission' from the Action Links and complete all additional questions in the submission questionnaire.

Once your full submission is complete, your paper will undergo a series of checks in preparation for peer review. After your manuscript has passed the checks we will send you a "revision decision", which will give you the opportunity to revise your manuscript as you have outlined in the current response to reviewers and then submit it to us. 

To provide the metadata for your submission, please Login to Editorial Manager (https://www.editorialmanager.com/pbiology) within two working days, i.e. by Mar 22 2023 11:59PM.

Kind regards,

Luke

Lucas Smith, Ph.D.

Associate Editor

PLOS Biology

lsmith@plos.org

---

## [Editor Report · Decision Letter 1]

27 Mar 2023

Dear Dr Zhou,

Thank you for completing the submission of your manuscript "Multi-omics analyses identify transcription factor interplay in corneal epithelial fate determination and disease" for consideration at PLOS Biology. As mentioned in our last email, your manuscript, the reviews from Review Commons, and your response to reviewers have been evaluated by the PLOS Biology editors and an Academic Editor with relevant expertise.

In light of the reviews from Review Commons, we will not be able to accept the current version of the manuscript as we agree with the reviewers that it would be important to validate the key claims in your study that are currently based on multiomics data. However, given our and the reviewers' interest in the study, we would welcome re-submission of a much-revised version that takes into account the reviewers' comments as outlined in your submission.

We cannot make any decision about publication until we have seen the fully revised manuscript and your response to the reviewers' comments. Your revised manuscript is also likely to be sent for further evaluation by the reviewers. We will do our best to engage with the same reviewers that initially reviewed the paper at Review Commons, however this is not always possible, as it depends on the reviewer availability. However, if we deviate from this commitment, we will let you know.

**IMPORTANT - SUBMITTING YOUR REVISION**

*Re-submission Checklist*

*Published Peer Review*

*PLOS Data Policy*

*Blot and Gel Data Policy*

Sincerely,

Luke

Lucas Smith, Ph.D.

Associate Editor

PLOS Biology

lsmith@plos.org

---

## [Decision Letter · Decision Letter 2]

23 Aug 2023

Dear Dr Zhou,

Thank you for your patience while we considered your revised manuscript "Multi-omics analyses identify transcription factor interplay in corneal epithelial fate determination and disease" for publication as a Research Article at PLOS Biology. This version of your manuscript, which has been revised in response to reviews from Review Commons, has been evaluated by the PLOS Biology editors, the Academic Editor and by the two original reviewers (reviewers 1 and 2). Additionally, after discussing the revision with the Academic Editor, we felt it was necessary to enlist the help of a third reviewer to provide additional expertise in corneal epithelial biology, to supplement the expertise of the reviewers chosen by Review Commons. 

The reviews are appended below. As you will see, both of the original reviewers are satisfied by the revision and they suggest we accept the manuscript. Reviewer 3 agrees that the manuscript represents a nice resource for the field, but has raised several concerns and suggestions to strengthen the paper. After discussing these comments with the Academic Editor, we think that Reviewer 3's comments can be addressed through additional discussion. While interesting, we would not require further analyses or experiments to address Reviewer 3's comments. We think you should, however, further elucidate the rationale for investigating corneal dystrophies and keratoconus in their study (related to FOSL2). You should also provide additional rationale for the inclusion of FOSL2 analysis in the context of glaucoma and FECD. If the content is not closely aligned with the manuscript's primary focus, you may consider shortening or omitting this.

We are likely to accept this manuscript for publication, provided you satisfactorily address these remaining points. 

**IMPORTANT: As you make these last changes, we also request that you address the following editorial requests: 

1) TITLE: We note that reviewer 1 has suggested some of the specific findings be added into the title. However, after discussing this comment within the team, we tend to lean towards leaving the title as more general. Instead, we might propose a minor tweak to make the title more declarative. 

If you agree, we suggest you change it to something like "Identification of the regulatory circuit governing corneal epithelial fate determination and disease". We will ultimately leave it up to you whether to change the title as we or Reviewer 1 suggests. 

2) DATA: Thank you for providing your data on GEO and EGA repositories and on UCSC genomebrowser trackhub. I suspect that this will largely meet our data sharing requirement which requires that all data be made available without restriction: https://journals.plos.org/plosbiology/s/data-availability

However I have a few queries regarding this data: 

a - I could not seem to access the GEO data, or find the EGA data to verify that it meets our requirements - can you provide me with a reviewer token to access this data? (sorry if I missed it somewhere)

b - I see that you mention that there will be controlled access to the EGA dataset - can you describe in more detail whether there will be restrictions to accessing this data, and why? Please provide detailed instructions for how readers can access that dataset. 

c - I searched the link you provided to the chipseq data on UCSC genome browser trackhub - but all I see is text (and I could not find the actual data). Can you point me to the dataset? (again, sorry if I missed it somewhere obvious). Also - are you able to generate a DOI for the data put on trackhub? If not, we kindly ask that you move the data to a repository where a DOI can be generated. 

d - As noted, the datasets provided largely meets the requirements of our data policy, which requires that all data be made available without restriction. However, for data presented in the study that is not included in the RNAseq and chipseq datasets, we ask that you provide a supplementary table containing the underlying data. Note that we do not require all raw data. Rather, we ask that all individual quantitative observations that underlie the data summarized in the figures. 

As an example, I saw that there was some qPCR data in supplemental figure 11. The data used to generate that figure would need to be provided as a supplemental table (as individual data points, not just the means). 

e - As a last request regarding the data - please update all of the figure legends, including supplemental, with a statement detailing where the underlying data can be found. For example, you can add the sentence "The data underlying this figure can be found at ____" (and then add details of the relevant repository, etc). 

For more info on our data policy (including allowable restrictions), see here: 

https://journals.plos.org/plosbiology/s/data-availability

We expect to receive your revised manuscript within two weeks. 

*Published Peer Review History*

*Press*

Sincerely,

Luke

Lucas Smith, Ph.D.

Senior Editor,

lsmith@plos.org,

PLOS Biology

Reviewer remarks:

Original Reviewer #1, Abhijeet Sonawane (note, this reviewer has signed their review): The authors have revised the manuscript substantially and have address the concerns and queries raised by this reviewer. This reviewer is very satisfied with the scientific quality of the manuscript and recommend its acceptance. I would think including PAX6 and FOSL2 relationship in title itself may increase the appeal of the paper.

Original Reviewer #2: In the revised manuscript, the authors included functional studies of the FOSL2, a gene predicted by their multi-omic analysis to play an essential role in corneal function. The functional validation included mapping the binding sites of FOSL2 by cut-and-run and knockdown approach to identify the targets. The authors also provide additional experimental evidence for regulating FOSL2 by PAX6. The authors addressed the reviewer's concerns on the data analysis and added new data that experimentally substantiates the role of FOSL2 in corneal function.

The study provides an overview of transcriptional regulators and targets in two essential tissues; human limbal stem cells and skin keratinocytes. The multi-omic comparison is valuable for delving into the processes governing cell differentiation and those involved in the disease mechanism of the cornea. Through their analyses and experimental validation, the authors expose the importance of FOXL2 in corneal function.

New Reviewer #3: This study generated multi-omics analysis of the corneal epithelium. The authors also compared the CREs between two stratified epithelia, corneal epithelial cells and keratocytes. They found a novel candidate, FOSL2, acts as a downstream transcription factor of PAX6. And this gene may associate with various eye diseases. These datasets offer an important resource for understanding the regulatory mechanism of corneal epithelial fate determination.

The major concern is that the potential function of FOSL2 in corneal epithelial cells found by the authors is related to angiogenesis and tight junction, however, the corneal epithelial diseases studied here is corneal dystrophies and keratoconus, the link between the phenotype and mechanism is pretty week. The focus for current manuscript is corneal epithelial cell fate, the analysis of FOSL2 in glaucoma, FECD, cataract, etc. seems unnecessary and lack of further evidence. The authors should identify the role of FOSL2 in real corneal epithelial diseases and verify the pathological changes. 

Additional comment：FOSL2 is expressed in both LSCs and KCs. Does FOSL2 function differently in KCs?

---

## [Editor Report · Decision Letter 3]

14 Sep 2023

Dear Dr Zhou,

Thank you for the submission of your revised Research Article "Identification of the regulatory circuit governing corneal epithelial fate determination and disease" for publication in PLOS Biology and for addressing the most recent editorial and reviewer requests in this revision. On behalf of my colleagues and the Academic Editor, Sui Wang, I am pleased to say that we are satisfied by the changes made and that we can, in principle, accept your manuscript for publication. Please note that before we can formally accept your manuscript and publish it, we will need you to address any remaining formatting and reporting issues which will be detailed in an email you should receive within 2-3 business days from our colleagues in the journal operations team; no action is required from you until then. 

**IMPORTANT: As you address any formatting and reporting requests, to come, please also address the following editorial request as well: 

1) Thank you for generating a new table with the qPCR data presented in your manuscript, and for referencing this dataset in the relevant figure legends. We understand that for other data presented in your study, that the underlying data is provided on trackhub, zenodo, and GEO and that this is clearly explained in the Data Availability Statement. We ask that you please also add a note to each figure legend (including supplemental) pointing readers to these datasets, where relevant. 

PRESS

Sincerely, 

Lucas Smith, Ph.D.,

Senior Editor

PLOS Biology

lsmith@plos.org